# DQPB: software for calculating disequilibrium U–Pb ages

**Timothy Pollard**[1,2], **Jon Woodhead**[1], **John Hellstrom**[1], **John Engel**[1], **Roger Powell**[1], and **Russell Drysdale**[1,2]

[1]School of Geography, Earth and Atmospheric Sciences, University of Melbourne, Melbourne, Victoria 3010, Australia
[2]EDYTEM UMR CNRS 5204, Université Savoie Mont Blanc, Le Bourget du Lac, CEDEX 73376, France

**Correspondence:** Timothy Pollard (pollard@student.unimelb.edu.au)

**Abstract.** Initial radioactive disequilibrium amongst intermediate nuclides of the U decay chains can have a significant impact on the accuracy of U–Pb ages, especially in young samples. For samples that can reasonably be assumed to have attained radioactive equilibrium at the time of analysis, a relatively straightforward correction may be applied. However, in younger materials where this assumption is unreasonable, it is necessary to replace the familiar U–Pb age equations with more complete expressions that account for growth and decay of intermediate nuclides through time. DQPB is software for calculating U–Pb ages while accounting for the effects of radioactive disequilibrium among intermediate nuclides of the U decay chains. The software is written in Python and distributed as both a pure Python package and a stand-alone graphical user interface (GUI) application that integrates with standard Microsoft Excel spreadsheets. The software implements disequilibrium U–Pb equations to compute ages using various approaches, including concordia intercept ages on a Tera–Wasserburg diagram, U–Pb isochron ages, Pb*/U ages based on single aliquots, and [207]Pb-corrected ages. While these age-calculation approaches are tailored toward young samples that cannot reasonably be assumed to have attained radioactive equilibrium at the time of analysis, they may also be applied to older materials where disequilibrium is no longer analytically resolvable. The software allows users to implement a variety of regression algorithms based on both classical and robust statistical approaches, compute weighted average ages and construct customisable, publication-ready plots of U–Pb age data. The regression and weighted average algorithms implemented in DQPB may also be applicable to other (i.e. non-U–Pb) geochronological datasets.

## 1 Introduction

With the exception of major uranium-bearing phases, rocks and minerals younger than a few million years were once considered virtually inaccessible to U–Pb methods owing to difficulties inherent in measuring the small quantities of radiogenic Pb generated over such short time periods (Getty and DePaolo, 1995). However, analytical advances over the past two decades, including improvements in pre-screening (Rasbury and Cole, 2009), sample preparation (e.g. Engel et al., 2020) and mass spectrometry (e.g. Getty and DePaolo, 1995; Woodhead et al., 2006; Sakata et al., 2014), have opened up the possibility of accurately and precisely dating materials as young as the Late Pleistocene age. These methodologies are now widely applied to radiogenic Pb-rich minerals including zircon (e.g. Paquette et al., 2019), as well as common Pb-rich materials such as carbonates (e.g. Richards et al., 1998), using both bulk and laser ablation (LA) or secondary ion mass spectrometry (SIMS) sampling techniques. In addition to analytical challenges in applying the U–Pb geochronometer to such young materials, another major issue lies in the need to accurately account for the effects of initial radioactive disequilibrium among intermediate nuclides of the U-series decay chains. For older samples, the effects of initial disequilibrium are often small relative to the precision of individual age determinations, but in younger materials, failure to correct for these effects can lead to large inaccuracies in final calculated ages (Ludwig, 1977; Schärer, 1984).

Secondary carbonates, such as speleothems, are well-known to be deposited out of radioactive equilibrium with respect to $^{234}U/^{238}U$, reflecting the $^{234}U/^{238}U$ ratios in the parent waters from which they form (Osmond and Cowart, 1992). Moreover, the insolubility of Th and Pa in these par-

ent waters leads to their near exclusion from newly precipitated carbonate, causing an additional component of disequilibrium (Richards et al., 1998). On the other hand, igneous minerals formed in high-temperature environments tend to be crystallised at, or very close to, radioactive equilibrium with respect to $^{234}$U/$^{238}$U, but out of equilibrium with respect to Th and Pa (Schoene, 2014). For example, minerals such as zircon tend to crystalise with $^{230}$Th/$^{238}$U ratios below radioactive equilibrium and initial $^{231}$Pa/$^{235}$U ratios in excess of radioactive equilibrium (Schmitt, 2007), whereas Th-rich phases, such as monazite, tend to crystalise with $^{230}$Th/$^{238}$U ratios in excess of radioactive equilibrium (Schärer, 1984). Over time, any initial excess or deficiency of intermediate nuclides gradually decreases as the U decay chains evolve toward radioactive equilibrium, eventually reaching a point after about 6–8 half-lives where disequilibrium effects are too small to be measured using current analytical techniques. For carbonates, this is typically 1.5 to 2 Ma for both $^{234}$U/$^{238}$U and $^{230}$Th/$^{238}$U, since evolution of $^{230}$Th toward equilibrium is constrained to follow that of the preceding nuclide $^{234}$U. For high-temperature minerals formed in equilibrium with respect to $^{234}$U/$^{238}$U but out of equilibrium with respect to $^{230}$Th/$^{238}$U, this age limit is typically closer to $\sim 0.5$ Ma.

There are two main approaches employed to account for the effects of radioactive disequilibrium on U–Pb ages. The first of these is applicable to samples that can reasonably be assumed to have attained radioactive equilibrium at the time of analysis. This involves correcting Pb*/U isotope ratios (where * denotes radiogenic Pb formed in situ by decay of U) for any excess or deficiency of intermediate nuclides relative to their radioactive equilibrium values (Schärer, 1984; Parrish, 1990). In a closed system, each daughter nuclide in initial excess or deficiency of equilibrium will cause an equivalent over or under abundance of Pb* once radioactive equilibrium is established (Mattinson, 1973). Therefore, it is possible to apply a relatively straightforward correction by adding or subtracting this excess or deficit of Pb*, provided the initial disequilibrium state is known or can be reliably estimated. Ages can then be computed using the regular U–Pb equations that disregard in-growth and decay of intermediate nuclides.

However, for younger samples, which cannot be assumed to be in a state of radioactive equilibrium at the time of analysis, it is necessary to replace the familiar U–Pb age equations with more complete expressions that can account for the growth and decay of intermediate nuclides through time. Equations of this form were first presented for the U–Pb system by Ludwig (1977) based on Bateman's (1910) general solution to differential equations that describes time evolution of radionuclides for an arbitrary linear decay chain. Later, Wendt and Carl (1985) presented an alternative version of these equations that includes some simplifying assumptions, whilst Guillong et al. (2014) provide a similar equation that accounts for disequilibrium in a single intermediate

nuclide only[1]. These "disequilibrium U–Pb" equations are general and can also be applied to older samples that have, in a practical sense, attained radioactive equilibrium at the time of analysis. On the other hand, inappropriate use of the Pb*-correction approach described above can lead to large over- or under-correction, and thus inaccuracy in calculated ages, over timescales similar to those in which analytically resolvable disequilibrium persists (Fig. 1).

As these more complete disequilibrium U–Pb equations are rather cumbersome to implement compared to the conventional U–Pb age equations, they are typically handled using specialised software or in-house computer code. Various approaches have been devised to achieve this. Isoplot (Ludwig, 2012) may still be the most widely used software in geochronology and contains built-in functions based on Ludwig (1977) that can be used to calculate disequilibrium U–Pb ages as part of a spreadsheet-based approach. However, this has a number of limitations. Firstly, Isoplot, which is distributed as an Excel add-in, is no longer being maintained and is incompatible with recent versions of Excel. Secondly, the Isoplot licensing status is ambiguous, and so it is unclear if the source code can be modified or extended, for example, to produce plots of disequilibrium U–Pb age data. Thirdly, numerical computing and plotting within the Excel environment is limited. More recently, other software packages for handling disequilibrium U–Pb age data have been developed (Engel et al., 2019), or are in the developmental stage (additions to the IsoplotR package of Vermeesch, 2018). However, this former solution runs on proprietary software that is not widely used in geochronology, and the latter is not yet documented within the peer-review literature and does not currently propagate disequilibrium correction uncertainties.

Here we introduce DQPB, a software package for calculating disequilibrium U–Pb ages. DQPB implements the disequilibrium U–Pb equations outlined below to compute ages using approaches that are suited to various young sample types. The following sections outline software functionality and discuss approaches that are implemented for age calculation, error propagation, linear regression, weighted average calculations and plotting.

## 2 Software overview

DQPB is written in Python, an interpreted, high-level, general-purpose programming language that is rapidly gaining popularity within the geosciences. DQPB is available as both a regular Python package and a stand-alone application that does not require users to have a separate Python distribution pre-installed (see Sect. 8 for further details). Python of-

---

[1]For the $^{238}$U–$^{206}$Pb decay series, the assumption inherent in this approach that $^{226}$Ra remains fixed at equilibrium with $^{230}$Th can lead to inaccuracy in the order of thousands of CEI years when [$^{230}$Th/$^{226}$Ra]$_i$ is significantly less than or greater than 1.

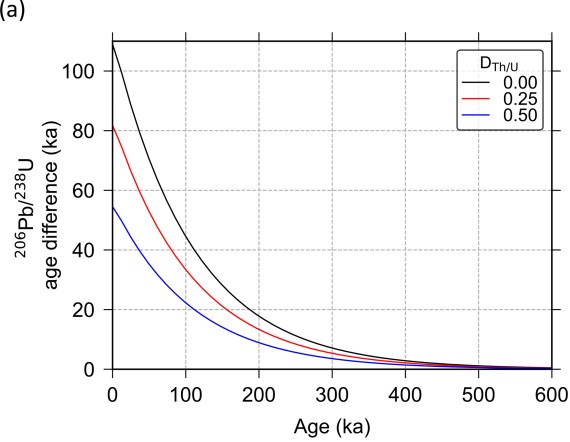

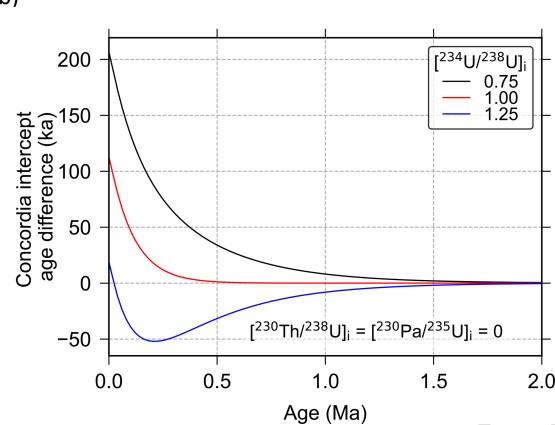

**Figure 1.** Comparison of U–Pb ages calculated using two different approaches: (i) ages corrected for disequilibrium assuming radioactive equilibrium has been established at the time of analysis and (ii) ages calculated using the more accurate disequilibrium U–Pb equations that account for growth and decay of intermediate nuclides through time (e.g. Eqs. 1 and 7). Age difference is given as age calculated via approach (i) (assumed equilibrium at time of analysis) minus age calculated via approach (ii) (more accurate approach). The top panel **(a)** shows a comparison of zircon $^{206}\text{Pb}/^{238}\text{U}$ ages calculated assuming various $D_{\text{Th/U}}$ values (where $D_{\text{Th/U}}$ denotes the ratio of mineral–melt partition coefficients). The bottom panel **(b)** shows a comparison of Tera–Wasserburg concordia intercept ages for carbonate samples assuming various initial $[^{234}\text{U}/^{238}\text{U}]$ values (where square brackets denote activity ratios).

fers several advantages as a language for scientific software development, including its open-source status, well-equipped libraries of functions and routines for scientific computing, and relatively easy-to-read syntax (e.g. Oliphant, 2007). Being a general-purpose language, Python also offers significant advantages in developing stand-alone graphical user interface (GUI) applications when compared to "domain-specific" scientific languages such as MATLAB and R.

DQPB is built on the core Python scientific computing libraries NumPy (Harris et al., 2020), SciPy (Virtanen et al., 2020) and Matplotlib (Hunter, 2007). It also takes ad-

vantage of PyQt to provide a modern GUI on macOS and Windows and xlwings to facilitate integration with Microsoft Excel. This allows users to select data from an open Excel spreadsheet, perform calculations via the graphical interface and have results (both numeric and figures) output to the same spreadsheet once computations are complete. In this way, it emulates the ease of use of the popular Isoplot programme (Ludwig, 2012). As is common practice with open-source software, all Python source code is available for viewing, download and modification via an online code repository (see Sect. 8).

## 3 Disequilibrium U–Pb age calculations

DQPB employs the equations of Ludwig (1977) to calculate U–Pb ages and plot disequilibrium age data. These equations were initially derived by Ludwig from a form of Bateman's (1910) solution that assumes zero initial abundance of all intermediate daughter nuclides and independently considers in-growth of Pb* from decay of the primordial parent and each preceding intermediate nuclide in the decay series (see also Ivanovich and Harmon, 1992; Neymark et al., 2000). These separate components are then summed, or "superposed" (Bateman, 1910), to obtain the total quantity of Pb* as a function of age, $t$.

Following this approach for the $^{238}\text{U}$ decay chain, and ignoring intermediate nuclides with a half-life less than or equal to that of $^{210}\text{Pb}$ (i.e. $\sim 22\,\text{a}$), results in an equation of the following form:

$$F = F_1 + F_2 + F_3 + F_4, \tag{1}$$

where $F = {}^{206}\text{Pb}^*/^{238}\text{U}$, and each term represents in-growth from the primordial parent (subscript 1) and initial abundances of each preceding intermediate daughter nuclide in the decay chain (subscripts > 1). In full, these individual components are

$$F_1 = e^{\lambda_{238}t} \left( c_1 e^{-\lambda_{238}t} + c_2 e^{-\lambda_{234}t} + c_3 e^{-\lambda_{230}t} \right.$$
$$\left. + c_4 e^{-\lambda_{226}t} 1 \right), \tag{2}$$

$$F_2 = \frac{\lambda_{238}}{\lambda_{234}} \left[ \frac{{}^{234}\text{U}}{{}^{238}\text{U}} \right]_i e^{\lambda_{238}t} \left( h_1 e^{-\lambda_{234}t} + h_2 e^{-\lambda_{230}t} \right.$$
$$\left. + h_3 e^{-\lambda_{226}t} + 1 \right), \tag{3}$$

$$F_3 = \frac{\lambda_{238}}{\lambda_{230}} \left[ \frac{{}^{230}\text{Th}}{{}^{238}\text{U}} \right]_i e^{\lambda_{238}t} \left( p_1 e^{-\lambda_{230}t} + p_2 e^{-\lambda_{226}t} + 1 \right), \tag{4}$$

$$F_4 = \frac{\lambda_{238}}{\lambda_{226}} \left[ \frac{{}^{226}\text{Ra}}{{}^{238}\text{U}} \right]_i e^{\lambda_{238}t} \left( 1 - e^{-\lambda_{226}t} \right), \tag{5}$$

where square brackets denote activity ratios; i denotes initial ratio; and $c$, $h$, and $p$ are Bateman coefficients given by Eq. (6) in Ludwig (1977), i.e.:

$$c_i/h_i/p_i = \frac{\prod_{j=1}^{n-1} \lambda_j}{\prod_{\substack{j=1 \\ i \neq j}}^{n} (\lambda_j - \lambda_i)}, \tag{6}$$

where $n$ is the number of nuclides in the part of the decay chain under consideration (this includes $^{206}$Pb, for which $\lambda = 0$, but excludes any preceding nuclides for $h$ and $p$). Similarly, for the $^{235}$U decay chain, we have

$$G = G_1 + G_2, \tag{7}$$

where $G = {}^{207}\text{Pb}^*/{}^{235}\text{U}$ and

$$G_1 = e^{\lambda_{235}t}\left(d_1 e^{-\lambda_{235}t} + d_2 e^{-\lambda_{231}t} + 1\right), \tag{8}$$

$$G_2 = \frac{\lambda_{235}}{\lambda_{231}}\left[\frac{{}^{231}\text{Pa}}{{}^{235}\text{U}}\right]_i e^{\lambda_{235}t}\left(1 - e^{-\lambda_{231}t}\right), \tag{9}$$

where $d$ is the Bateman coefficient defined in an equivalent manner to above. Identical equations may also be derived via the matrix exponential approach (e.g. Albarède, 1995), or using the Laplace transformation (Catchen, 1984). However, we have opted to preserve the original Bateman form for the purpose of clarity and because we see no advantage in adopting these alternative forms here. These disequilibrium U–Pb equations may be employed to compute ages using single aliquot or diagrammatic approaches in a similar fashion to the more familiar U–Pb equations, although they require numerical methods to solve in all instances (see discussion below).

When dealing with materials young enough to retain $[^{234}\text{U}/^{238}\text{U}]$ or $[^{230}\text{Th}/^{238}\text{U}]$ values that are analytically resolvable from radioactive equilibrium, it is generally more accurate to use present-day (i.e. measured) activity ratios rather than assumed initial values. This information can be incorporated into the above equations by employing an "inverted" form of the U-series age equations, whereby initial activity ratios are expressed as a function of present-day ratios and $t$ (Woodhead et al., 2006). These equations may then be substituted into the disequilibrium U–Pb equations above and included in the numerical solving procedure, resulting in a solution to both age and the initial activity ratio value. For example, this approach has been widely applied to Quaternary speleothems using measured $[^{234}\text{U}/^{238}\text{U}]$ values (e.g. Woodhead et al., 2006; Pickering et al., 2011; Bajo et al., 2012).

## 3.1 Pb*/U and $^{207}$Pb-corrected ages

The most straightforward implementation of the disequilibrium U–Pb age equations outlined above involves treating each U decay series independently to compute a radiogenic $^{206}$Pb/$^{238}$U or $^{207}$Pb/$^{235}$U age. This is achieved by solving

$$\left(\frac{{}^{206}\text{Pb}^*}{{}^{238}\text{U}}\right)_{\text{meas.}} - F = 0, \tag{10}$$

or

$$\left(\frac{{}^{207}\text{Pb}^*}{{}^{235}\text{U}}\right)_{\text{meas.}} - G = 0, \tag{11}$$

where the subscript meas. denotes a measured ratio corrected for blank and common Pb, and $F$ and $G$ are given above (Eqs. 1–5, and 7–9). This age-calculation approach may be applied, for example, to compute $^{206}$Pb*/$^{238}$U ages in young, radiogenic Pb-rich minerals such as Quaternary zircons, provided that common Pb is negligible or can be accurately corrected for (e.g. von Quadt et al., 2014).

Where common Pb is not negligible, or not amenable to accurate correction based on the measurement of $^{204}$Pb-based ratios (e.g. in samples analysed by ICP–MS techniques), a version of the $^{207}$Pb-corrected age employed by SIMS analysts (e.g. Williams, 1998), but modified to account for disequilibrium (Sakata, 2018), may be more practically useful. This approach, which is similar to the "single-aliquot" method of Woodhead et al. (2012) for calculating ages in high U/Pb speleothems, involves plotting each data point, uncorrected for common Pb and disequilibrium, on a Tera–Wasserburg diagram ($^{207}$Pb/$^{206}$Pb vs. $^{238}$U/$^{206}$Pb; Tera and Wasserburg, 1972) and projecting a line from a common initial $^{207}$Pb/$^{206}$Pb value on the $y$-axis intercept, through each data point to the disequilibrium concordia (Fig. 2). An intercept age may then be computed for each data point, assuming concordance between the $^{238}$U and $^{235}$U decay schemes (Chew et al., 2011). This provides a means of correcting ages for common Pb and disequilibrium in an internally consistent fashion. However, unlike the disequilibrium concordia intercept approach outlined below (Sect. 3.3), the common Pb composition is not given by linear regression of the data points themselves and must be specified independently. For igneous minerals, this may be achieved using whole rock measurements, analysis of Pb isotope ratios in co-genetic phases with high common Pb/U ratios (e.g. K-feldspars) or model estimates of average crustal Pb composition, such as that of Stacey and Kramers (1975).

To compute disequilibrium U–Pb ages using these approaches, it is necessary to specify the initial radioactive disequilibrium state of long-lived intermediate nuclides. For minerals that are assumed to have crystallised from a melt in secular equilibrium, $[^{230}\text{Th}/^{238}\text{U}]_i$ may be computed according to the following relationship (e.g. McLean et al., 2011):

$$\left[\frac{{}^{230}\text{Th}}{{}^{238}\text{U}}\right]_i = \frac{(\text{Th/U})_{\text{min.}}}{(\text{Th/U})_{\text{melt}}} = D_{\text{Th/U}}, \tag{12}$$

where min. denotes mineral and $D_{\text{Th/U}}$ is the ratio of mineral–melt partition coefficients (i.e. $D_{\text{Th}}/D_{\text{U}}$). An equivalent expression may be written for $[^{231}\text{Pa}/^{235}\text{U}]_i$. Based on this relationship, it is possible to account for disequilibrium in computing U–Pb ages for co-genetic igneous minerals using one of two different approaches, each entailing different assumptions regarding mineral–melt partitioning.

Approach (i) assumes that the Th/U elemental ratio of the melt is constant, but may vary across different grains. For this approach, Th/U$_{\text{melt}}$ is estimated from whole rock measurements (Schärer, 1984), or measured Th/U in co-genetic phases assumed to be representative of the original

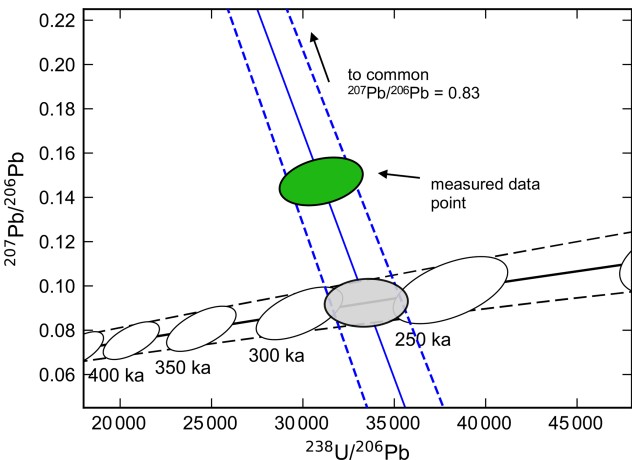

**Figure 2.** Graphical representation of a $^{207}$Pb-corrected age calculation. A straight line (blue) is projected from the initial $^{207}$Pb/$^{206}$Pb value at the $y$-axis intercept through the measured data point to the disequilibrium concordia curve, constructed here using $D_{Th/U} = 0.2 \pm 0.1$ ($2\sigma$) and $D_{Pa/U} = 2.9 \pm 0.8$ ($2\sigma$). The dashed black lines represent uncertainties (95 % CI) in the trajectory of the concordia arising due to distribution coefficient uncertainties (see Sect. 6). Age markers along the disequilibrium concordia are shown as 95 % confidence ellipses (white), also accounting for distribution coefficient uncertainties. The $^{207}$Pb-corrected age of $275 \pm 23$ ka (95 % CI) is represented by the grey intercept ellipse (95 % confidence). Note that the equilibrium concordia, if plotted, would appear as a horizontal line along the bottom of this figure at $y \approx 0.046$.

melt composition (e.g. volcanic glasses, Rioux et al., 2012). Th/U$_{min.}$ values are then determined based on either direct measurement of $^{232}$Th/$^{238}$U in each mineral grain (e.g. in LA-ICP-MS analyses, Guillong et al., 2014), or inferred from the radiogenic $^{208}$Pb/$^{206}$Pb ratio and age using an iterative procedure (e.g. in TIMS analyses, Crowley et al., 2007). Approach (ii), on the other hand, assumes that $D_{Th/U}$ is constant for all mineral grains, implying that Th/U of the magma may be heterogenous. For this approach, $D_{Th/U}$ values may be estimated based on experimental values or average values from geologically similar contexts (e.g. Sakata, 2018). For estimating $[^{231}$Pa/$^{235}$U$]_i$ values, the second approach is more widely applicable, owing to difficulties inherent in constraining Pa/U values of the melt. It is also more easily justified for $^{207}$Pb-corrected ages owing to their lower sensitivity to this value (Sakata et al., 2017).

For a suite of co-genetic mineral grains that are thought to belong to a single statistical population, a weighted average age may be computed using equivalent approaches to conventional U–Pb ages. However, in the case of disequilibrium ages, uncertainty in Th/U$_{melt}$ for approach (i) outlined above, or $D_{Th/U}$ and $D_{Pa/U}$ for approach (ii), acts as a systemic component of uncertainty, giving rise to correlated age uncertainties. These correlations can be non-trivial and should be considered in any weighted average calculation to accurately propagate assigned uncertainties and avoid artifi-

cially deflating mean squared weighted deviation (MSWD) statistic (McLean et al., 2011). DQPB allows users to compute disequilibrium $^{206}$Pb*/$^{238}$U and $^{207}$Pb-corrected ages, specifying the initial disequilibrium state using either of the approaches above. For approach (i), either a measured $^{232}$Th/$^{238}$U or radiogenic $^{208}$Pb/$^{206}$Pb ratio with analytical uncertainty is input for each aliquot along with a common Th/U$_{melt}$ value and uncertainty. For approach (ii), a common $D_{Th/U}$ ($D_{Pa/U}$) value and uncertainty is input and applied to all aliquots under the assumption that these uncertainties are perfectly correlated. Age uncertainties and uncertainty covariances are then estimated by either Monte Carlo methods or analytical uncertainty propagation (see Sect. 5), and where appropriate, weighted average ages accounting for this covariance structure may be computed using either classical or robust statistical approaches (see Sect. 4 for further details).

## 3.2 "Classical" U–Pb isochron ages

Disequilibrium $^{238}$U–$^{206}$Pb and $^{235}$U–$^{207}$Pb "classical" isochron ages may be computed for common Pb-rich samples by numerically solving $F - b = 0$ or $G - b = 0$, where $b$ is the slope of the isochron regression line on a $^{206}$Pb/$^{204}$Pb vs. $^{238}$U/$^{204}$Pb or $^{207}$Pb/$^{204}$Pb vs. $^{235}$U/$^{204}$Pb diagram, respectively. For classical U–Pb isochron diagrams, isotope ratios are traditionally referenced to $^{204}$Pb, however, when dating young materials with very low $^{232}$Th abundance, such as carbonates with low detrital content, it is also possible to reference to $^{208}$Pb instead under the assumption that $^{232}$Th has produced negligible radiogenic $^{208}$Pb since the time of system closure (Getty et al., 2001). The two formulations are mathematically equivalent, but the latter can be advantageous where accurate an measurement of $^{204}$Pb proves difficult, such as in ICP–MS dating of young samples (Engel et al., 2019). While U–Pb isochron approaches can be less reliable than concordia intercept ages (Ludwig, 1998), especially for young datasets incorporating the low abundance $^{204}$Pb isotope, they are offered in DQPB because of their potential utility in computing ages for Pb-rich materials where the disequilibrium state of only one of the U-series decay chains is well constrained.

## 3.3 Concordia intercept ages

Concordia intercept ages are well-suited to Pb-rich materials such as carbonates and apatite that typically contain variable Pb*/common Pb ratios within individual growth horizons (Woodhead and Pickering, 2012; Chew et al., 2011; Engel and Pickering, 2022). To compute ages using this approach, multiple co-genetic samples uncorrected for common Pb are plotted on a Tera–Wasserburg diagram. If all samples (i) have remained closed to the exchange of U-series isotopes post crystallisation, (ii) contain varying quantities of common Pb with an identical $^{207}$Pb/$^{206}$Pb composition

and (iii) were initially crystallised in the same disequilibrium state, they form a mixing line on a Tera–Wasserburg diagram between a purely radiogenic end-member lying on the concordia curve (the locus of all radiogenic Pb ICs through time) and a common Pb end-member at the $y$-axis intercept (Tera and Wasserburg, 1972). When accounting for the effects of radioactive disequilibrium, the familiar equilibrium concordia is replaced with a family of disequilibrium concordia constructs (e.g. Wendt and Carl, 1985), based on the following equations:

$$x = \frac{^{238}\mathrm{U}}{^{206}\mathrm{Pb}^*} = \frac{1}{F}, \tag{13}$$

and

$$y = \frac{^{207}\mathrm{Pb}^*}{^{206}\mathrm{Pb}^*} = U^{-1}Gx, \tag{14}$$

where $U$ denotes the present-day natural $^{238}\mathrm{U}/^{235}\mathrm{U}$ ratio. Activity ratios may either be input directly into functions $F$ and $G$ as initial values, or as present-day values via the inverted U-series equations as described in Sect. 3, and ages are then calculated as the intersection of a regression line with the appropriate concordia curve, by solving

$$U^{-1}G - aF - b = 0, \tag{15}$$

where $a$ and $b$ are the slope and $y$-intercept values obtained by linear regression of the data points. `DQPB` allows users to fit a variety of regression models to Tera–Wasserburg data (Sect. 4), compute ages based on either initial or present-day (i.e. measured) intermediate nuclide activity ratios values, and construct customisable plots of the disequilibrium concordia intercept ages (e.g. Fig. 3).

## 3.4 "Forced concordance" initial [$^{234}\mathrm{U}/^{238}\mathrm{U}$] values

`DQPB` also implements a version of the "forced concordance" routine of Engel et al. (2019), which targets closed-system samples where the initial $^{234}\mathrm{U}/^{238}\mathrm{U}$ activity ratio is unknown, but activity ratios of other long-lived intermediate nuclides (i.e. [$^{230}\mathrm{Th}/^{238}\mathrm{U}$] and [$^{231}\mathrm{Pa}/^{235}\mathrm{U}$]) are reliably constrained (e.g. very low initial Th carbonates). The routine determines the [$^{234}\mathrm{U}/^{238}\mathrm{U}$] value that forces concordance between the $^{235}\mathrm{U}$–$^{207}\mathrm{Pb}$ and $^{238}\mathrm{U}$–$^{206}\mathrm{Pb}$ decay schemes based on individual U–Pb isochrons and outputs this value along with its uncertainty computed by Monte Carlo methods. This algorithm may be useful for characterising initial [$^{234}\mathrm{U}/^{238}\mathrm{U}$] values for particular geological contexts (e.g. cave sites when dating carbonate speleothems) where all available samples lie beyond the range of measurable disequilibrium.

## 4 Linear regression and weighted average age protocols

Linear regression and weighted average age algorithms capable of accounting for analytical uncertainties and accom-

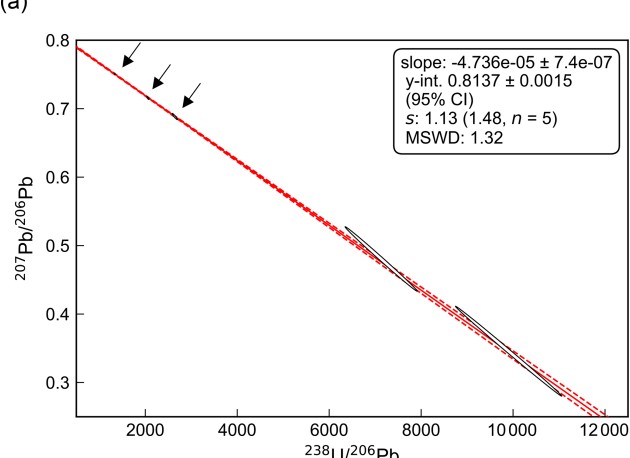

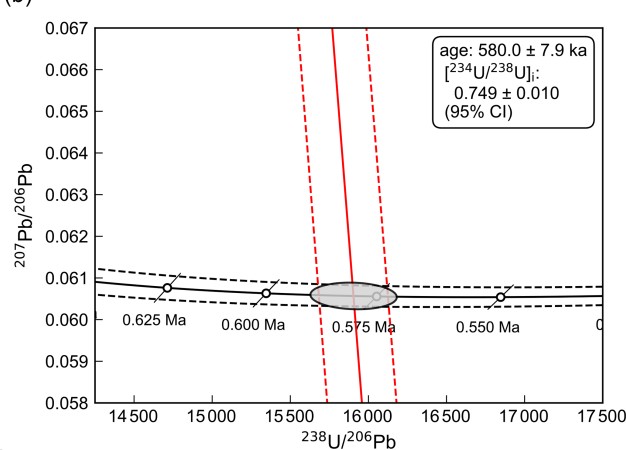

**Figure 3.** Example of Tera–Wasserburg concordia intercept age plots for Middle Pleistocene stalagmite CCB (see Sect. 7.1 for further details). **(a)** Plot showing the `spine` linear regression fit to data (red line), with dashed red lines indicating uncertainties (95 % confidence). Measured data points (white 95 % confidence ellipses), exhibit a strong negative uncertainty correlation due the effects of blank subtraction (Woodhead et al., 2012). **(b)** Enlarged view of the concordia intercept. The disequilibrium concordia line (black) is constructed using a measured [$^{234}\mathrm{U}/^{238}\mathrm{U}$] value of $0.9512 \pm 0.0013$ ($2\sigma$), with initial activity ratios for other intermediate nuclides assumed equal to 0. The dashed black lines indicate uncertainties (95 % confidence) arising from uncertainty in this measured [$^{234}\mathrm{U}/^{238}\mathrm{U}$] value. Regular concordia age markers are shown as white circles, and the black diagonal lines represent 95 % confidence age ellipses, which are collapsed to straight line segments because there is no uncertainty assigned to [$^{231}\mathrm{Pa}/^{235}\mathrm{U}$] (see Sect. 6). The grey "intercept ellipse" (95 % confidence) is representative of the $10^6$ simulated concordia intercept points from the Monte Carlo simulation.

modating the possibility of "excess scatter" (i.e. scatter in excess of that which is attributable to assigned analytical uncertainties) are crucial for attaining reliable U–Pb ages. `DQPB` offers two distinct approaches to perform linear regression and compute weighted averages. The first of these (Sect. 4.1)

is rooted in a classical statistical paradigm and emulates the default protocols of `Isoplot` (Ludwig, 2012). The second approach (Sect. 4.2) takes advantage of recent developments in the application of robust statistics to geochronology, implementing the `spine` algorithm of Powell et al. (2020) as well a weighted average variant of this algorithm (the "spine" weighted average), and a newly developed robust regression algorithm (the "robust model 2"). Although users are free to choose the most appropriate algorithm for their particular dataset, the `spine` linear regression and weighted average algorithms are set as the default because they are considered suitable for a wider range of datasets than their classical statistics-based counterparts.

## 4.1 Classical statistical approaches

For the classical statistics-based approach, linear regression and weighted averaging of data are first performed using algorithms that weight data points according to assigned analytical uncertainties under the assumption that these are the only source of data point scatter. For linear regression, this involves implementing the error-weighted least-squares algorithm of York et al. (2004), which yields equivalent results to the original algorithm of York (1969) with uncertainties on regression parameters calculated following the maximum likelihood estimation (MLE) approach of Titterington and Halliday (1979). For weighted average calculations, an uncertainty-weighted least-squares algorithm is implemented whereby individual data points are weighted according to the inverse of their analytical variance, accounting for the uncertainty covariance structure where relevant (e.g. McLean et al., 2011). An apparent advantage of this classical approach is that it allows a statistic with a well-established distribution, i.e. the MSWD, to be used to assess data point scatter in relation to measurement uncertainties under the assumption that residuals are strictly Gaussian distributed (Wendt and Carl, 1985). Probabilistic-based conclusions can then be drawn regarding the likely presence (or not) of excess scatter.

Where the MSWD lies within a probabilistically acceptable range about 1, as indicated either by a confidence interval on MSWD (e.g. Powell et al., 2002) or, equivalently, the "probability-of-fit" value (Ludwig, 2012), the initial least-squares solution and analytical uncertainty-based standard errors are retained. However, if the MSWD value falls outside such limits, the dataset is deemed likely to contain a component of excess scatter, which may be either "geological scatter" (variability in initial Pb composition, open-system behaviour etc.) or some component of analytical uncertainty that is unaccounted for. Provided that the MSWD is not unreasonably high, assigned analytical uncertainties likely still dominate the uncertainty budget, and, on this basis, the initial least-squares solution is retained, but uncertainties are inflated so as to reduce the MSWD to 1. For linear regression fitting, this may be termed the "model 1x", bor-

rowing the terminology of Powell et al. (2020). On the other hand, where the MSWD lies well outside a probabilistically acceptable range, the assumptions of the `York` fit or analytical uncertainty-weighted average are clearly violated, and it is commonplace to either manually reject data points to restore scatter to an acceptable range or turn to alternative classical statistics-based approaches, for example, by employing the `Isoplot` model 2 or model 3 fits (Ludwig, 2012).

Although this classical statistics-based approach is predominant within geochronology, it has some limitations. Firstly, the rejection of outliers from small sample sizes typical in geochronology is notoriously difficult. Secondly, this approach relies on a stepwise mode of uncertainty handling, which is both conceptually unsatisfying and requires the choice of arbitrary cut-off points, the values of which can have a substantial impact on calculated ages and uncertainties (see Powell et al., 2020). Thirdly, the MSWD statistic is very sensitive to small departures in residuals from a strict Gaussian distribution, making it an overly sensitive indicator of excess scatter for many real-world geochronological datasets, which are often slightly "heavy tailed" (Rock et al., 1987; Powell et al., 2002). And lastly, the model 2 and 3 linear regression algorithms are not well-suited to all datasets. For example, the model 3 fit parameterises excess scatter as an external Gaussian-distributed component of scatter, an assumption that is difficult to justify in the typical case where the precise cause of excess scatter is not well-established nor known to be strictly Gaussian (Ludwig, 2003). The model 2 fit, on the other hand, makes few assumptions regarding the statistical distribution of the excess scatter, however, it weights all data points equally and does not account for analytical uncertainties at all.

## 4.2 Robust statistical approaches

Robust algorithms, which do not rely on the assumption of Gaussian-distributed residuals, offer a means of addressing some of the limitations of the classical statistics-based approach outlined above. Robust statistical approaches have previously been proposed in geochronology, including the median-of-medians linear regression algorithm (Siegel, 1982), which is implemented in `Isoplot` (Ludwig, 2012) and weighted average algorithms of varying complexity (e.g. Rock et al., 1987; Ludwig, 2012). While these algorithms are resistant to the effects of outliers, a limitation of these approaches is that they ignore analytical uncertainties, leading to suboptimal results where these do in fact constitute a significant component of the total data point scatter. The `spine` linear regression algorithm (Powell et al., 2020) improves on these previous robust approaches by accounting for assigned analytical uncertainties; it also exhibits a number of favourable properties that arguably make it more generally applicable to geochronological datasets compared to the classical statistics-based approach.

The `spine` algorithm minimises a piece-wise objective function (the "Huber loss function"), whereby data points lying along a central linear band (i.e. the spine) are given full weighting, but points falling outside this band are progressively down-weighted according to their weighted residual. Uncertainties on regression parameters are calculated using a first-order error propagation approach and tend to increase smoothly with increasing data point scatter. Notably, in the case where all data points lie within this central band, `spine` yields identical results to `York`, making this algorithm suitable for both "well-behaved" and excess scatter datasets, provided that the majority of data points comprise a well-defined linear array within their uncertainties. In the place of the MSWD, a robust metric called the spine width, $s$, is used to assess whether or not data point scatter is commensurate with accurate use of this algorithm given assigned uncertainties. The $s$ term `CE2` is the median absolute deviation of weighted residuals, normalised to be equal to the standard deviation for a strictly Gaussian distribution (i.e. NMAD). This statistic tends toward 1 for well-behaved datasets and may be used in a similar manner to the MSWD, although, in contrast to MSWD, confidence intervals on $s$ must be derived from simulation rather than from a formal statistical distribution (Powell et al., 2020). `DQPB` outputs $s$ along with this simulated upper 95 % confidence bound (here denoted $s_{lim}$) allowing users to assess if the central spine of data is sufficiently well-defined for use of this regression algorithm.

For computing robust weighted averages, `DQPB` also offers a one-dimensional variant of the `spine` linear regression algorithm, termed the `spine` weighted average (see Appendix A). The `spine` weighted average is capable of accounting for assigned analytical uncertainties, and like the classical least-squares approach, it can accommodate uncertainty correlations among data points. Analogously to the linear regression version, it gives full analytical weighting to data at the centre of the distribution, and progressively down-weights data points lying away from this central spine according to the Huber loss function. In the case where data point scatter is commensurate with analytical uncertainties, the `spine` weighted average reduces to the classical statistics weighted mean (e.g. Powell and Holland, 1988; McLean et al., 2011). Equivalently to the `spine` regression algorithm, the quality of this central spine of data points can be assessed by considering $s$ in relation to $s_{lim}$ which is derived via the simulation of Gaussian distributed datasets (see Appendix B).

In addition to the `spine` linear regression algorithm, `DQPB` also offers a second robust linear regression approach for datasets that have an $s$ value exceeding $s_{lim}$ but are still reasonably thought to have age significance (see Appendix C). This regression algorithm, named the "Robust model 2", is similar to the `Isoplot` model 2, but encompasses robust properties which reduce the influence of outliers on the fitted line in a similar manner to `spine`. Although this algorithm discards analytical uncertainties and

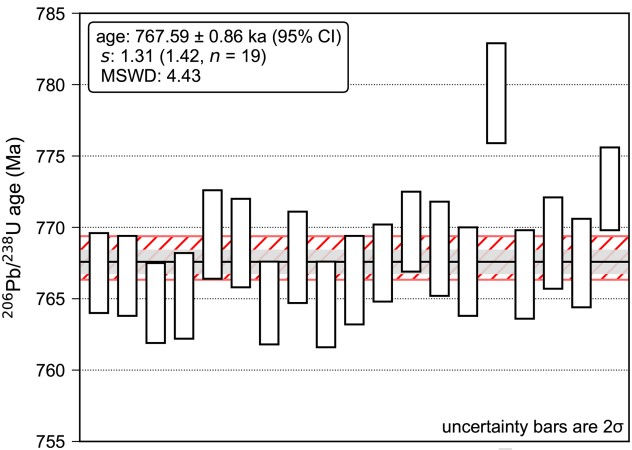

**Figure 4.** A comparison of the `spine` weighted average with the classical statistics weighted mean for Bishop Tuff zircon [206]Pb/[238]U ages from Crowley et al. (2007). The black line shows the `spine` weighted average [206]Pb/[238]U age of $767.59 \pm 0.86$ ka (95 % CI), with this uncertainty indicated by the light grey shading. The spine width value, $s$, for this dataset is 1.31, which is within the upper 95 % confidence limit of $s$ ($s_{lim} = 1.42$, $n = 19$), suggesting that the dataset contains a sufficiently well-defined spine of data points for use of this algorithm. For comparison, the classical statistics weighted mean age is $767.85 \pm 1.5$ ka (95 % CI) (MSWD = 4.43, $n = 19$), with this uncertainty represented by the hatched red area. If the two oldest ages are treated as outliers, as for the preferred age in the original publication, the classical statistics weighted mean shifts to $767.06 \pm 0.85$ ka (95 % CI) (MSWD = 1.3, $n = 17$). Note that age uncertainty covariances have not been considered in this example, although the `spine` algorithm is capable of accounting for these where necessary (see Sect. 4).

provides less reliable age uncertainty estimates than `spine`, it is offered as a robust alternative to the model 2 and 3 fits discussed above, as it is expected to be applicable to a wider range of datasets.

## 5   Age uncertainty propagation

First-order analytical uncertainty propagation is a suitable method for computing U–Pb age uncertainties in cases where input variables (e.g. Pb/U measurements, regression fit parameters and activity ratios) have relatively small uncertainties, and the age equation is linear with respect to these variables within the neighbourhood of the age solution (e.g. Barlow, 1989). However, this approach can be inaccurate where uncertainties on input variables are relatively large, and the linear approximation breaks down. For example, applying analytical uncertainty propagation to isochron or concordia intercept ages with large regression fitting uncertainties can result in inaccurate age uncertainties, because the age solution PDF (probability density function) can be markedly asymmetric.

Monte Carlo simulation is an alternative approach to propagating U–Pb age uncertainties, which is not reliant on a linear approximation. With this approach, input variables are randomly sampled from within their PDFs (typical Gaussian or multivariate Gaussian distributions) for each trial, and an age is calculated. This procedure is then repeated many times to build up an estimate of the output age PDF from which summary statistics (e.g. standard deviation or confidence intervals) can be estimated. Monte Carlo simulation is capable of accounting for asymmetric age distributions and can provide accurate results even when uncertainties on input variables are large (e.g. Albarède, 1995). However, a potential drawback of the Monte Carlo approach is that, owing to its stochastic nature, it requires a large number of trials to produce numerically accurate and stable results, making it more computationally intensive than analytical approaches. The reliability of Monte Carlo uncertainty estimates scales with the number of trials. Although the number of trials required to produce 95 % confidence intervals that are accurate to about two significant digits depends on the shape of the output age PDF, $10^5$–$10^6$ trials are typically sufficient (e.g. JCGM, 2008). For most age calculations in `DQPB`, Monte Carlo uncertainty calculations involving $10^5$–$10^6$ trials are completed in a matter of seconds, although, calculations can take longer in cases where either uncertainties on input variables are very large, there is a high proportion of trials with non-convergent age solutions, and/or a large number of single-aliquot ages are computed simultaneously. `DQPB` allows users to adjust the number of Monte Carlo trials. By default, this is set to $5 \times 10^4$ for convenience, but it is recommended to increase this to $\sim 10^6$ when computing final age uncertainties by Monte Carlo simulation.

## 5.1 Monte Carlo uncertainties

`DQPB` offers Monte Carlo uncertainty propagation for all disequilibrium U–Pb age types. For concordia intercept and classical isochron datasets fitted either using robust or model 1 algorithms, regression parameters are randomised within uncertainties according to a multivariate Gaussian distribution for each trial, accounting for uncertainty correlation between the slope and $y$-intercept. For model 1x, model 2 and model 3 fits, (i.e. excess scatter fits) regression parameters are instead randomised within their "observed scatter" uncertainties, i.e. $1\sigma$ analytical uncertainties multiplied by $\sqrt{\mathrm{MSWD}}$ according to a bivariate $t$ distribution with $n-2$ degrees of freedom, where $n$ is the number of data points. Activity ratios, either as initial or present-day values, are then randomised according to univariate Gaussian distribution, and an age is computed for each combination of inputs. In cases where a present-day activity ratio value is given, the initial activity ratio value is also computed for each trial as part of the numerical solving procedure. Age uncertainties are reported as a 95 % confidence interval, estimated from the 2.5 and 97.5 percentiles of simulated ages.

The application of Monte Carlo uncertainty propagation to disequilibrium ages, computed using present-day (i.e. measured) activity ratio values that are not clearly resolvable from radioactive equilibrium (i.e. where the activity ratio PDF significantly overlaps the radioactive equilibrium value), can produce unreliable results. This is because random samples drawn from the overlapping part of the measured activity ratio PDF tend to produce non-convergent age solutions, and this may bias the output age distribution. To address this issue, `DQPB` performs two checks to verify that the input data are suitable for Monte Carlo uncertainty propagation. The first check is performed prior to commencing the simulation and ensures that the measured activity ratio values are analytically resolvable from equilibrium with 95 % confidence. Where this criterion is not met, a warning is displayed to the user, and the Monte Carlo simulation does not proceed. The age is still reported, but the uncertainties are listed as undefined. The second check, which is performed after a Monte Carlo simulation is completed, verifies that a minimum number of trials were successful (the default value is set to 97.5 %). Where this second criterion is not met, the software displays a warning that Monte Carlo simulation results may be unreliable and should not be used. This second warning may also be triggered if the PDF of an initial activity ratio significantly overlaps negative values (e.g. if the value of an initial activity ratio is assigned a value close to zero with some uncertainty), which may also lead to unreliable age uncertainty estimates.

For multiple co-genetic Pb*/U and $^{207}$Pb-corrected ages, an approach similar to Renne et al. (2010) is used to account for systematic components of uncertainty. With this approach, isotope ratios for each data point are first randomised within their individual analytical uncertainties according to a Gaussian distribution (or a multivariate Gaussian distribution for $^{207}$Pb-corrected ages). Variables that contribute a systematic component of uncertainty, such as distribution coefficients or Th/U$_{\mathrm{melt}}$ ratios (and initial $^{207}$Pb/$^{206}$Pb values for $^{207}$Pb-corrected ages) are then randomised within their uncertainties once per trial, and this common value is used to compute an age for each data point. This procedure results in an $m$-by-$n$ array of simulated ages (where $m$ is the number of Monte Carlo trials and $n$ is the number of single-aliquot ages) displaying the covariance resulting from their common dependence on these variables (e.g. Renne et al., 2010). Age uncertainties on individual aliquots are reported as 95 % confidence intervals and age covariances are estimated from simulated ages for each of the $n$ data points, resulting in an $n$-by-$n$ age covariance matrix. Where appropriate, this covariance structure is then used in subsequent weighted average age calculations.

## 5.2 Analytical uncertainty propagation

In addition to Monte Carlo uncertainty propagation, `DQPB` offers first-order analytical uncertainty propagation for

Pb*/U and [207]Pb-corrected ages. While Monte Carlo methods can provide more accurate age uncertainties for these age types when uncertainties on input variables are large and result in asymmetric age distributions (as discussed above), such asymmetries are not typically accounted for in computing weighted averages. Although it is possible in principle to account for the effects of asymmetries in weighted averages, such approaches are not yet well developed in geochronology. At the same time, the computation time required to implement Monte Carlo uncertainty propagation for large $n$ (number of aliquots) and large $m$ (number of trials) can be much more significant than for diagrammatic ages. For these reasons, analytical uncertainty propagation may be preferable for these age types, provided that uncertainties on input variables are relatively small, and/or age uncertainties on all data points are known to be approximately Gaussian. `DQPB` implements a matrix-based approach for analytical uncertainty propagation that accounts for the effect of random and systematic components of uncertainty on each aliquot and keeps track of all covariance terms (e.g. McLean et al., 2011). This approach allows the age covariance structure to be easily computed and used in subsequent weighted average age calculations in an equivalent manner to the Monte Carlo approach discussed above.

## 6    Data visualisation and plotting

`DQPB` outputs customisable plots for all diagrammatic and weighted average U–Pb age calculations. For isochron and concordia ages, a plot of the linear regression fit is provided, showing data points as 95 % confidence ellipses along with the regression line and a 95 % confidence band on the regression fit. This confidence band is plotted using the approach of Ludwig (1980) for model 1, 2 and 3 fits and Monte Carlo simulation for robust fits (e.g. Fig. 3a). For concordia intercept ages, an additional plot is also provided, showing an enlarged view of the intersection between the isochron and the disequilibrium concordia curve (e.g. Fig. 3b). The intercept points of all Monte Carlo simulated ages are also shown on this plot, either as $m$ $x-y$ points or plotted as a single 95 % confidence ellipse representing the population of simulated intercept points. For [207]Pb-corrected ages, data points are plotted on a Tera–Wasserburg diagram. If $D_{\text{Th/U}}$ and $D_{\text{Pa/U}}$ are input as constant values for all data points, a disequilibrium concordia curve may also be plotted along with projection lines from the common Pb point through each data point to its concordia intercept (e.g. Fig. 5a).

For disequilibrium concordia curves on concordia intercept plots, age markers may be plotted as either point markers or "age ellipses" that represent uncertainty in $x-y$ for a given $t$ value arising from uncertainty in activity ratio values. Where there is uncertainty in activity ratios for both the [238]U and [235]U decay series, these age ellipse markers are true ellipses, akin to those representing decay constant uncertainties on an equilibrium Tera–Wasserburg concordia diagram (Ludwig, 1998). On the other hand, where there is activity ratio uncertainty assigned to only one of the U decay schemes, these age ellipses collapse to line segments with a slope equivalent to the Tera–Wasserburg isochron lines described in Eq. (7) of (Wendt and Carl, 1985). A 95 % confidence band representing uncertainty in the trajectory of the concordia curve arising from uncertainty in activity ratios may also be plotted, based on a Monte Carlo simulation. `DQPB` allows users to customise a wide range of plot settings, export figures in a variety of image file formats and access all numeric data used to construct plots via output to a new Excel spreadsheet (see Supplement for further details).

## 7    `DQPB` usage examples

### 7.1    Concordia intercept speleothem age

Despite their relatively low U content, clean (i.e. with low detrital content) carbonates, such as speleothems, can be well-suited to U–Pb dating provided they contain relatively high U/Pb ratios and spread in U/Pb ratios within individual growth layers (Woodhead et al., 2012). Here, we demonstrate computation of a concordia intercept age for a Middle Pleistocene speleothem CCB from Corchia Cave, Italy, based on solution MC-ICP-MS analyses. The sample is young enough to retain a $[^{234}\text{U}/^{238}\text{U}]$ ratio which is analytically resolvable from equilibrium but lies just beyond reach of the [230]Th geochronometer using routine methods. A measured $^{234}\text{U}/^{238}\text{U}$ activity ratio of $0.9512 \pm 0.0013$ ($2\sigma$) was used in the age calculation, obtained via MC-ICP-MS (Hellstrom, 2003). Speleothems from this cave site consistently exhibit very low detrital Th (as reflected in $^{232}\text{Th}/^{230}\text{Th}$ ratios; Drysdale et al., 2012) and thus the initial $[^{230}\text{Th}/^{238}\text{U}]$ is assumed equal to 0. The initial activity ratios for other intermediate nuclides are likewise assumed equal to 0. The data are regressed using the `spine` algorithm, which in this case returns equivalent results to the `York` algorithm (Fig. 3a). A lower intercept age of $580 \pm 7.9$ ka (95 % CI) is computed, along with an initial $[^{234}\text{U}/^{238}\text{U}]$ value of $0.749 \pm 0.010$ (95 % CI). Age uncertainties are estimated by Monte Carlo simulation using $10^6$ trials (Fig. 3b).

### 7.2    Quaternary zircon [207]Pb-corrected ages

In this example, we demonstrate a [207]Pb-corrected age calculation for a suite of zircons from the Sambe–Kisuki tephra (Shuhei Sakata, unpublished data), which is thought to have erupted approximately 100 ka ago from the Sambe volcano located in the Shimane prefecture in the west of Japan. Analyses were performed by multi-collector LA-ICP-MS using a method similar to Hattori et al. (2017). Disequilibrium ages were calculated following approach (i) outlined in Sect. 3.1 using a $D_{\text{Th/U}}$ value of $0.2 \pm 0.03$ TS1 ($2\sigma$), a $D_{\text{Pa/U}}$ value of $2.9 \pm 1.0$ ($2\sigma$) and a common $^{207}\text{Pb}/^{206}\text{Pb}$ value based on

the two-stage model of Stacey and Kramers (1975). With this approach, uncertainties in $D_{Th/U}$ and $D_{Pa/U}$ are propagated as purely systematic components of uncertainty. Age uncertainties were calculated by the Monte Carlo simulation, using $10^6$ trials for each age point (Fig. 5a). These uncertainties are identical (within the quoted number of significant figures) to those obtained by analytical uncertainty propagation.

Computing a weighted average using a classical statistics-based approach (accounting for uncertainty correlations), gives a weighted mean age of $96.6 \pm 39$ ka (95 % CI), with an MSWD of 3.54, indicating a very high probability of excess scatter in the dataset under the assumption of Gaussian-distributed residuals. On the other hand, the robust `spine` weighted average algorithm gives a weighted average age of $94.2 \pm 10.9$ ka (95 % CI) (Fig. 5b), with a $s$ value of 1.28 which lies within the upper 95 % confidence limit of $s$ ($s_{lim} = 1.57$, $n = 6$). This suggests that there is a sufficiently well-defined spine of data at the centre of the distribution for use of this algorithm, and thus the weighted average is likely to carry age significance under the assumption that crystallisation of these zircons constitutes a geologically discrete event (e.g. see Ickert et al., 2015). Note that the `spine` weighted average algorithm down-weights the single point lying away from the average age line, and thus it has little influence on the computed weighted average. For comparison, excluding this point gives a classical weighted average age of $92.1 \pm 6.3$ ka (95 % CI) with an MSWD of 0.55.

## 8 Availability and distribution

`DQPB` (Pollard, 2023a, https://doi.org/10.5281/zenodo.7804190) is released under a MIT license, permitting modification of the source code and re-distribution with minimal restrictions. The source code may be viewed via an online code repository (see: https://github.com/timpol/DQPB, last access: 6 April 2023). This repository also contains links to downloadable installers for macOS and Windows, as well as online documentation. Suggestions for bug fixes and new features, as well as pull requests, are also accepted via this repository.

In addition to the stand-alone GUI version of the software, `DQPB` is also available as part of a pure Python package named `pysoplot` (Pollard, 2023b, https://doi.org/10.5281/zenodo.7804162), offering greater flexibility for more experienced Python users. The `pysoplot` package is hosted at a separate online repository (see: https://github.com/timpol/pysoplot, last access: 6 April 2023) and is available via pip (the package installer for Python) – see: https://pypi.org/project/pysoplot/ (last access: 6 April 2023).

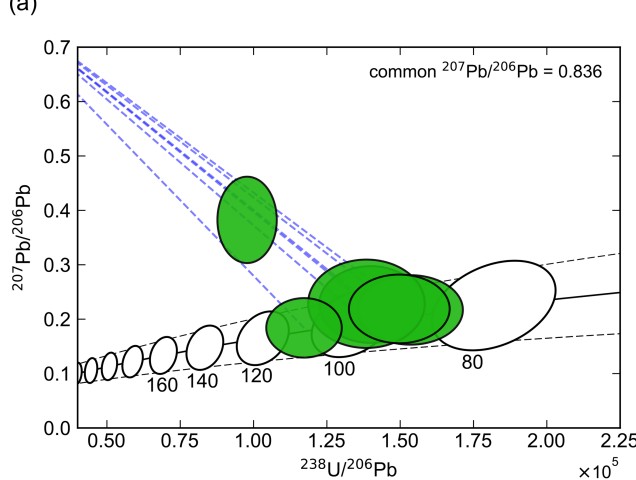

(a)

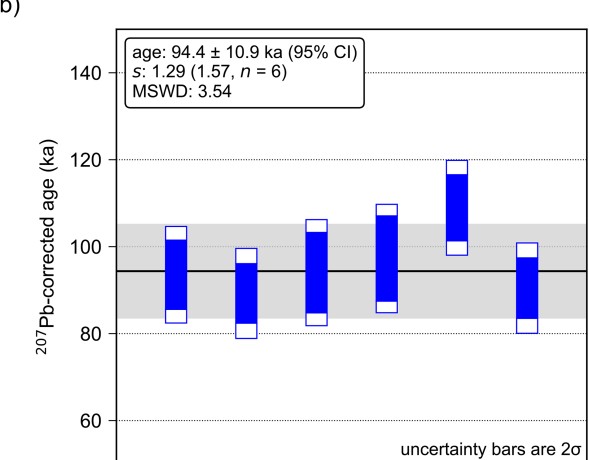

(b)

**Figure 5.** Example of $^{207}$Pb-corrected age plots. **(a)** Data ellipses plotted on a Tera–Wasserburg diagram as 95 % confidence ellipses (green). The black line shows the disequilibrium concordia constructed using $D_{Th/U} = 0.2 \pm 0.03$ (2$\sigma$) and $D_{Pa/U} = 2.9 \pm 1.0$ (2$\sigma$), with dashed black lines indicating uncertainty bounds (95 % CI). Concordia markers are plotted as 95 % age ellipses, representing $x-y$ uncertainty for a given age due to uncertainty in distribution coefficients. The dashed blue lines show a line projecting from the common Pb point at the $y$ intercept ($^{207}$Pb/$^{206}$Pb = 0.836) through the centre of each data point to its intercept with the disequilibrium concordia. **(b)** Plot of individual $^{207}$Pb-corrected ages. The dark blue bars indicate age uncertainties (2$\sigma$) accounting for random analytical uncertainties only, while the larger white bars show combined random and systematic uncertainties (2$\sigma$) (i.e. including components due to uncertainty in $D_{Th/U}$ and $D_{Pa/U}$ values). The black line shows the weighted average age computed using the robust `spine` algorithm which accounts for the age covariance structure, with the light grey shading indicating a 95 % confidence interval on this weighted average. Initial disequilibrium corrections were applied assuming constant distribution coefficient ratios, i.e. approach (i) in Sect. 3.1, treating uncertainties in $D_{Th/U}$ and $D_{Pa/U}$ as purely systematic uncertainties.

## 9   Conclusion

This paper introduces `DQPB`, an open-source software package for calculating disequilibrium U–Pb ages. The software implements disequilibrium U–Pb equations to compute ages using various approaches, including disequilibrium Pb*/U ages based on single aliquots, U–Pb isochron ages and concordia intercept ages on a Tera–Wasserburg diagram. Various linear regression and weighted average age algorithms are implemented in the software, including those based on both classical and robust statistics, and high-quality "publication-ready" figures are output. A key feature of the stand-alone GUI-based version of the software is that it allows close integration with Microsoft Excel and thus continues the legacy of `Isoplot` in allowing straightforward interaction with U–Pb datasets from within a simple spreadsheet environment. `DQPB` is free open-source software, and all source-code is available for viewing and download via an online repository. For more experienced Python users, `DQPB` is available as part of a pure Python package, which may be downloaded and modified with minimal restrictions to meet individual requirements. This software will continue to be developed under an open-source model and new features will be added in the future.

## Appendix A: `spine` robust weighted average

Following the logic of Powell et al. (2020) for the two-dimensional case, a robust spine CE3 weighted average accounting for analytical uncertainties may be obtained for one-dimensional data (e.g. multiple coeval ages). To achieve this in the general case where correlated age uncertainties are permitted, it is first necessary to express weighted residuals in an uncorrelated form. In the classical statistics-based solution (e.g. Powell and Holland, 1988; McLean et al., 2011), the weighted average age is obtained by finding $\bar{t}$ that minimises the sum of squared weighted residuals:

$$S = (t - \bar{t}\mathbf{1})\mathbf{V}_t^{-1}(t - \bar{t}\mathbf{1}), \tag{A1}$$

where $t$ is a column vector of ages, $\mathbf{1}$ is a column vector of ones, and $\mathbf{V}_t$ is the uncertainty (covariance) matrix of the ages. To apply the Huber loss function, which is defined as

$$\rho(r_k) = \begin{cases} r_k^2 & \text{if } |r_k| \le h \\ 2hr_k - h^2 & |r_k| > h, \end{cases} \tag{A2}$$

where $r_k$ is the weighted residual of the $k$th data point and $h = 1.4$, the sum of weighted residuals must first be recast as a sum of uncorrelated weighted residuals. This may be achieved via eigendecomposition of the covariance matrix:

$$\mathbf{V}_t = \mathbf{Q}\mathbf{\Lambda}\mathbf{Q}^\mathrm{T}, \tag{A3}$$

where $\mathbf{\Lambda}$ is the eigenvalue matrix consisting of positive eigenvalues on the diagonals and $\mathbf{Q}$ is the eigenvector matrix. From this we obtain

$$\mathbf{V}_t^{-1/2} = \mathbf{Q}\mathbf{\Lambda}^{-1/2}\mathbf{Q}^\mathrm{T}, \tag{A4}$$

which can be substituted into Eq. (A1) to give

$$S = r^\mathrm{T}r, \tag{A5}$$

where $r$ is a column vector of weighted residuals, given by

$$r = \mathbf{V}_t^{-1/2}(t - \bar{t}\mathbf{1}). \tag{A6}$$

Following the approach in Powell et al. (2020), we minimise $\sum \rho_k$ by finding the $\bar{t}$ value that solves

$$\mathbf{1}^\mathrm{T}\mathbf{V}_t^{-1/2}\psi(r) = 0, \tag{A7}$$

where

$$2\psi = \frac{\partial \rho}{\partial r_k}. \tag{A8}$$

This is achieved using an iterative reweighting procedure, whereby the weight function $w(r_k) = \psi(r_k)/r_k$ is introduced, resulting in

$$\mathbf{1}^\mathrm{T}\mathbf{W}e = 0, \tag{A9}$$

with

$$e = t - \bar{t} \tag{A10}$$

and

$$\mathbf{W} = \mathbf{W}_h\mathbf{V}_t^{-1}, \tag{A11}$$

such that $\mathbf{W}_h$ is a diagonal matrix having $w(r_k)$ as the $kk$th element. This combines the weighting from $w$ with the weighting from the correlated uncertainties on $t$. Rearranging this gives an expression equivalent to Eq. (B13) in Powell et al. (2020):

$$\bar{t} = \mathbf{1}^\mathrm{T}\mathbf{W}t(\mathbf{1}^\mathrm{T}\mathbf{W}\mathbf{1})^{-1}, \tag{A12}$$

which can be solved by iteration from a robust starting point (e.g. Maronna, 2019). Analogous to the development of Eq. (B17) in Powell et al. (2020), uncertainties on $\bar{t}$ are then computed by first-order uncertainty propagation as

$$\sigma_{\bar{t}} = \frac{1}{\sqrt{\mathbf{1}^\mathrm{T}\mathbf{V}_t^{-1}\mathbf{I}'\mathbf{1}}}, \tag{A13}$$

where $\mathbf{I}' = \mathrm{diag}(\dot{\psi}(r))$.

In the case where all $|r_k| < h$, then $\psi(r) = r$, $\mathbf{W}_h = \mathbf{V}_t^{-1}$ and $\mathbf{I}' = \mathbf{I}$, so

$$\bar{t} = \mathbf{1}^\mathrm{T}\mathbf{V}_t^{-1}t\left(\mathbf{1}^\mathrm{T}\mathbf{V}_t^{-1}\mathbf{1}\right)^{-1} \tag{A14}$$

and

$$\sigma_{\bar{t}} = \frac{1}{\sqrt{\mathbf{1}^\mathrm{T}\mathbf{V}_t^{-1}\mathbf{1}}}, \tag{A15}$$

yielding the classical statistics-based result.

**Table B1.** Simulated 95 % confidence intervals for $\sqrt{\text{MSWD}}$ and $s$ (spine width) as a function of the number of data points, $n$, where * denotes a one-sided upper 95 % confidence limit. The results for MSWD are equivalent to those obtained from formal statistical tables. DQPB outputs the one-sided upper 95 % confidence limit on s (denoted $s_{\text{lim}}$ in the software) to evaluate suitability of the spine weighted average algorithm for a particular dataset. DQPB also outputs equivalent values for the spine linear regression which are given in Table 1 of Powell et al. (2020).

| $n$ | $\sqrt{\text{MSWD}}$ | | | $s$ | | |
|---|---|---|---|---|---|---|
| | Low | High | * | Low | High | * |
| 5 | 0.348 | 1.669 | 1.540 | 0.12 | 1.94 | 1.72 |
| 7 | 0.454 | 1.552 | 1.449 | 0.22 | 1.83 | 1.65 |
| 9 | 0.522 | 1.481 | 1.392 | 0.29 | 1.74 | 1.59 |
| 15 | 0.634 | 1.366 | 1.301 | 0.43 | 1.59 | 1.47 |
| 29 | 0.739 | 1.260 | 1.215 | 0.58 | 1.42 | 1.34 |
| 59 | 0.818 | 1.181 | 1.151 | 0.70 | 1.30 | 1.24 |
| 6 | 0.408 | 1.602 | 1.488 | 0.21 | 1.75 | 1.57 |
| 8 | 0.491 | 1.513 | 1.412 | 0.29 | 1.70 | 1.55 |
| 10 | 0.548 | 1.454 | 1.371 | 0.35 | 1.65 | 1.52 |
| 16 | 0.646 | 1.354 | 1.291 | 0.46 | 1.54 | 1.44 |
| 30 | 0.744 | 1.256 | 1.211 | 0.60 | 1.41 | 1.33 |
| 60 | 0.820 | 1.180 | 1.149 | 0.71 | 1.29 | 1.24 |

## Appendix B: spine weighted average *s* simulations

To assess whether the central spine of data points is sufficiently well-defined to obtain a meaningful weighted average, we compare the spine width, $s$, to its upper 95 % confidence limit bound derived via simulation of Gaussian-distributed datasets. Simulations were performed using sample sizes, $n$, ranging between 5–100 data points. For each $n$, $10^6$ pseudorandom samples were drawn from a standard normal distribution; $s$ values were computed for each sample, and confidence limits on $s$ were estimated based on relevant percentiles (see Table B1). Odd and even $n$ are considered independently in order to account for the effect of small sample bias inherent to NMAD (e.g. Hayes et al., 2014). The impact of different uncertainty covariance structures on $s$ were also examined and found to have a negligible effect on these confidence limits.

## Appendix C: Robust model 2

The robust data-fitting algorithm in Powell et al. (2020) in the two-dimensional case, and above in Appendix A, in the one-dimensional case, are predicated on the one-sided confidence intervals on the spine width (in Table 1, last column of Powell et al., 2020), and in Table B1 here). The calculation of age, and particularly the uncertainty on age, is appropriate for the case where a dataset gives a spine width that is consistent with the confidence interval.

Not covered is how best to proceed if in fact a dataset is not consistent with the confidence interval. Whereas the argument developed in Powell et al. (2020), and by extension here, is that datasets which are consistent with this interval are likely to have age significance, this becomes progressively more awkward to argue as the spine width increases. The view taken in this section is that the calculations advocated are for datasets that are considered to have age significance, commonly by geological inference, even though the spine width is outside the confidence interval.

Once the spine width is too large, the data-fitting should plausibly not depend on the analytical uncertainties on the data as these are deemed insufficient to account for the observed scatter. A clear-cut and robust way to proceed is then to discard the analytical uncertainties and rely on the scatter of the data – specifically the spine width – to provide the data uncertainties.

Model 2 in Isoplot provides a framework for how to proceed. As outlined in the Appendix of Powell et al. (2020), for the Isoplot model 2, in which analytical uncertainties are discarded, data are fit $y$ on $x$, and $x$ on $y$, and the results combined, circumventing the potentially deleterious effects of error-in-variables effects (e.g. Fuller, 1987). In Isoplot, such calculations are done by applying ordinary least squares in the two calculations, giving the slopes, $b_{yx}$ and $1/b_{xy}$, respectively, with the combined slope being given by

$$b = \pm\sqrt{b_{yx}b_{xy}} = \pm\sqrt{\frac{\sum(y_k - \overline{y})^2}{\sum(x_k - \overline{x})^2}} \tag{C1}$$

and

$$a = \overline{y} - b\overline{x} \tag{C2}$$

(see Powell et al., 2020, for notation and details).

In the equivalent of model 2 using the spine algorithm, the analytical uncertainties are discarded, then the spine width is calculated from the scatter of the data about the line, $s = \text{nmad}(e)$. The development in Appendix B of Powell et al. (2020) can be applied as is to the two calculations required: $y$ on $x$ and $x$ on $y$, except that the two definitions need to be changed: Eq. (B5)[2] should involve $\mathbf{W}_e$ with diagonal elements, $1/s$, and Eq. (B13) should involve $\mathbf{W}$ with diagonal elements, $w(r_k)/s^2$.

Applying the spine algorithm in the above-modified form to fitting $y$ on $x$ and $x$ on $y$ allows the slope $b = \pm\sqrt{b_{yx}b_{xy}}$ and the intercept $a$ to be calculated, as in Appendix A3 of Powell et al. (2020). The covariance matrix for each slope and intercept can be calculated by Eq. (B17). Combination into a covariance matrix for $\{a, b\}$ requires the observation that $b_{yx}$ and $b_{xy}$ are uncorrelated. An error propagation is then straightforward to $b$, and in fact a good approximation is generally given by adding the constituent covariance matrices and dividing by 4.

---

[2]The equation numbers here refer to those in Powell et al. (2020).

**Code availability.** The `DQPB` source code and compiled versions of the GUI application may be obtained from the repositories described in Sect. 8.

**Data availability.** The example datasets discussed in Sect. 7 are included in the Supplement.

**Supplement.** The supplement related to this article is available online at: https://doi.org/10.5194/gchron-5-1-2023-supplement.

**Author contributions.** JW and JH devised the initial concept with input from JE and TP. TP devised the computational approaches and wrote the software with assistance from JE and JH. RP and TP devised the spine CE4 weighted average approach. RP devised the robust model 2 approach and wrote the code for it. TP wrote the paper with contributions from all co-authors.

**Competing interests.** The contact author has declared that none of the authors has any competing interests.

**Disclaimer.** Publisher's note: Copernicus Publications remains neutral with regard to jurisdictional claims in published maps and institutional affiliations.

**Acknowledgements.** Shuhei Sakata kindly provided the unpublished Sambe-Kisuki zircon data. We thank reviewers Pieter Vermeesch and Ryan Ickert, as well as the associate editor Noah M. McLean, for their carefully considered comments and suggestions that have greatly improved the paper.

**Financial support.** This research was supported by Australian Research Council grants DP110102185 (to Russell Drysdale, Jon Woodhead and John Hellstrom), DP160102969 (to Russell Drysdale and Jon Woodhead) and FL160100028 (to Jon Woodhead).

**Review statement.** This paper was edited by Noah M. McLean and reviewed by Ryan Ickert and Pieter Vermeesch.

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

## Remarks from the language copy-editor

## Remarks from the typesetter