# Peer review of "DQPB v0.1.0 USER GUIDE"

_Geochronology, 2022_

## Referee Comment (RC1)

**Review of "DQPB: software for calculating disequilibrium U-Pb ages" by T. J. Pollard *et al.*'**

Pieter Vermeesch
University College London
`p.vermeesch@ucl.ac.uk`

October 28, 2022

This paper presents methods and software to correct U–Pb data for U-series disequilibrium. The methods are based on the classic Bateman (1908) equations, whereas the software combines `Python` with `Excel`. As the author of the open source `IsoplotR` package for geochronological data processing, I applaud the authors' decision to share their source code on `GitHub`. I was also excited to to see that DQPB is part of a software suite called `pysoplot`. It would be great if the geochronology community had multiple, independently developed `Isoplot` alternatives, as this would stimulate further innovation and allow users to cross-check results. DQPB is easy to install on `Windows` and `Mac OS`, but not on `Linux`. This is, presumably, because DQPB's graphical user interface (GUI) interacts with `Excel`, which is not available on `Linux`.

DQPB's online documentation is detailed and extensive, but only covers the GUI. It would be useful if it also covered the command-line API. Not only would this allow `Linux` users to access DQPB, but it would also benefit power users on other operating systems. For example, in the second half of this review, I will carry out some numerical experiments using `R`. Had there been a DQPB API, then I would have been able to use `Python` instead, which would have been a more efficient way to make my points.

I tested DQPB on a `Windows` machine and found the software easy to install and straightforward to use, even without reading the instructions. The integration with `Excel` reminded me why `Isoplot` was so successful: it makes a lot of sense to group the data processing tool together with the data itself, in a spreadsheet. I haven't really been able to replicate this experience in `IsoplotR`. One slightly annoying issue is that DQPB overprints the data with the results. This problem probably only occurs on `Windows` (DQPB's lead developer appears to use `Mac OS`) and should be easy to fix.

I tested DQPB on a number of samples and got similar results to `IsoplotR`. This is not surprising given that the two programs use, essentially, the same equations, although `IsoplotR` casts them in a matrix form. The manuscript is

a bit dismissive of `IsoplotR`'s disequilibrium corrections, even though these are more extensive than `DQPB`'s current capabilities and include 3-dimensional 'Total U-Pb' isochron regression and Ludwig (1998)-style error propagation, neither of which are implemented in `DQPB`. However, there are two things that `DQPB` does, and which `IsoplotR` doesn't.

The first of these is robust isochron regression as an alternative to `IsoplotR`'s model-3 regression, using the spine algorithm of Powell et al. (2020). I have no objections to or comments on this. The second difference is that `IsoplotR` currently does not propagate the uncertainty of the initial disequilibrium correction. The reason for this limitation is that there is no easy analytical solution to the error propagation problem, and numerical solutions are too slow for a software package that can be run online. `DQPB` estimates the uncertainty of the disequilibrium correction by Monte Carlo simulation. I believe that this is a similar approach to that taken by `Isoplot`. The Monte Carlo approach works as follows:

1. Given a linear array of isotopic data in Tera-Wasserburg (i.e. $^{238}$U/$^{206}$Pb vs. $^{207}$Pb/$^{206}$Pb) space, and the measured $^{234}$U/$^{238}$U activity ratio ($\mu_{4/8}$) and its uncertainty ($\sigma_{4/8}$);

2. Draw a random value ($[4/8]_m$, say) from a normal distribution with mean $\mu_{4/8}$ and standard deviation $\sigma_{4/8}$;

3. Carry out a linear regression through the data and find the initial $^{234}$U/$^{238}$U activity ratio ($[4/8]_i$) and isochron age ($t$) that is consistent with both the U–Pb data and $[4/8]_m$;

4. Repeat steps 2 and 3 until the entire distribution of measured $^{234}$U/$^{238}$U activity ratios has been sampled;

5. If step 3 fails, or produces physically impossible results (e.g., $t < 0$), then ignore the corresponding $[4/8]_m$. Otherwise add the $[4/8]_i$ and $t$-values to a list of a acceptable results;

6. Use the spread of the acceptable $[4/8]_i$ and $t$-values to quantify their respective uncertainties.

Since `DQPB` does not work on my computer, I implemented my own version of this algorithm, using `R` and `IsoplotR`. The only major difference between my code and `DQPB` is that it does not sample the $[4/8]_m$-distribution randomly, but uses a targeted approach to sample $[4/8]_m$ as a sequence of regularly spaced normal quantiles. This has two advantages. First, it requires orders of magnitude fewer iterations (50 vs. 30,000). Second, it produces a deterministic result, unlike the Monte Carlo approach, whose results depend on the seed of a random number generator. To verify that the `R` code produces equivalent results to the `Python` code, Figure 1 analyses the 'Corchia' dataset from the manuscript. The results are, essentially, the same as those that I obtained using `DQPB` on `Windows`:

[Figure]

Figure 1: Output of a `DQPB`-like algorithm for the Corchia dataset. a) Tera-Wasserburg concordia diagram with disequilibrium-corrected isochron (age=0.5803 Ma); b) 50 representative samples from the $^{234}U/^{238}U$ measurement distribution; c) The corresponding initial $^{234}U/^{238}U$ activity ratios; d) The isochron ages corresponding to the initial $^{234}U/^{238}U$ activity ratios presented in panel c.

Next, let us apply the same approach to a much older sample, such as the 'Hoogland' data set of Pickering et al. (2019):

| [38/06] | $2\sigma\%[38/06]$ | [07/06] | $2\sigma\%[07/06]$ | $\rho$ |
|---|---|---|---|---|
| 843.8 | 7.59420 | 0.0697 | 0.00742305 | -0.9991 |
| 843.9 | 4.64145 | 0.0673 | 0.00450910 | -0.9972 |
| 737.4 | 6.26790 | 0.1671 | 0.00618270 | -0.9982 |
| 834.6 | 10.43250 | 0.0749 | 0.01048600 | -0.9994 |
| 845.5 | 9.72325 | 0.0696 | 0.00967440 | -0.9995 |
| 781.6 | 5.47120 | 0.1189 | 0.00552885 | -0.9984 |
| 787.1 | 6.69035 | 0.1212 | 0.00672660 | -0.9578 |
| 834.5 | 6.25875 | 0.0719 | 0.00611150 | -0.9976 |

The uncorrected U–Pb isochron age for this sample is 7.407 Ma, which is 30 half–lives of $^{234}$U. Consequently, the measured present-day $^{234}$U/$^{238}$U activity ratio ($[4/8]_m$) is statistically indistinguishable from secular equilibrium, at $1.0016 \pm 0.001$ (1se). However, when we apply the DQPB approach to this data set:

[Figure]

Figure 2: Output of the `DQPB`-like algorithm for the Hoogland dataset of Pickering et al. (2019). Panels a)-d) are as in Figure 1. The black dots in panel b) mark synthetic replicates that are rejected because they yield physically impossible $[4/8]_i$ and/or $t$-values.

Despite the lack of measurable disequilibrium, the `DQPB` approach appears to have successfully applied a disequilibrium correction, resulting in a corrected age that is less than half the uncorrected age. Although the magnitude of the correction is huge, the precision of the corrected age appears to be surprisingly good, at less than 15%. How is this possible? The answer lies in the rejected solutions (step 5 of the algorithm), which are marked in black in Figure 2.b.

To demonstrate that the result of Figure 2 is wrong, let us replace the measured $^{234}\text{U}/^{238}\text{U}$ activity ratio with the equilibrium ratio. Thus

$$\mu_{4/8} = \frac{\lambda_{234}}{\lambda_{234} - \lambda_{238}} = 1.000055(\pm 0.001)$$

Plugging this value into the `DQPB` algorithm yields the following result:

[Figure]

Figure 3: The same data as Figure 2, but assuming perfect equilibrium of the measured $^{234}$U/$^{238}$U activity ratio. Note that half of the synthetic replicates have been rejected (black circles). Even though there is absolutely no evidence for disequilibrium, the DQPB approach nevertheless seems very confident that the isochron age should be halved compared to the uncorrected data.

Once again, it appears that the algorithm has achieved the impossible, and has corrected the disequilibrium effect in the absence of any detectable deviation from the equilibrium ratio. It has done so by ignoring 50% of the $[4/8]_m$ distribution. This is because, for this old sample, essentially any $[4/8]_m$ value that is less than the equilibrium ratio would require a negative $[4/8]_i$ ratio, or a negative isochron age $t$.

In fairness to DQPB, the program does report the number of failed inversions. I would argue that solutions that include rejected tries should not be trusted. It would be better to replace the ad-hoc Monte Carlo approach with a more rigorous alternative. I believe that there are two approaches for doing this. The first option is to use the traditional ('frequentist') approach, which uses the method of maximum likelihood. I have attempted this but got stuck: the problem quickly becomes too complex for a non-statistician. The second option is to use a Bayesian approach. This is much easier to implement, as I will demonstrate in a basic form:

1. Define a prior distribution for $[4/8]_i$. In the following example, I will use a uniform distribution from $m$ to $M$ (e.g., $m = 0$ and $M = 20$ for $[4/8]_i$-values running from 0 to 20), but this could easily be replaced by a more informative prior.

2. Draw a random sample from this prior distribution, carry out the constrained isochron regression and report the resulting age ($t$) and remaining $[4/8]_m$.

3. Calculate the likelihood of the inferred $[4/8]_m$ values under a normal distribution with mean $\mu_{4/8}$ and standard deviation $\sigma_{4/8}$.

4. Repeat steps 2 and 3 and combine the likelihood values with the prior distribution to produce a 'posterior' distribution of $[4/8]_i$ values and the correponding age distributions. This can either be done using a Markov chain, or with a targeted approach of regularly spaced $[4/8]_i$ values.

Applying this approach to the Corchia example yields essentially identical results to the DQDB results of Figure 1:

[Figure]

Figure 4: The same data as Figure 1, but using the Bayesian approach. For this sample, the DQPB method yields similar results to the Bayesian solution.

However, when the Bayesian approach is applied to the Hoogland data, it produces a very different, and I would argue more sensible, result than DQPB:

[Figure]

Figure 5: Application of the Bayesian approach to the Hoogland data of Pickering et al. (2019). The maximum posterior likelihood agrees with the mode of the `DQPB` solution of Figure 2. However, whereas `DQPB` suggests a high degree of confidence in the disequilibrium correction, the Bayesian result shows that one cannot rule out a much older age. In fact, the uncorrected U–Pb age of 7.4 Ma is almost as likely as the corrected age of 3.3 Ma.

Note how the posterior distribution still has a maximum at $t = 3.3$ Ma and $[4/8]_i = 12$, just like the `DQPB` solution of Figure 2. But unlike the `DQPB` solution, the Bayesian solution assigns a significant likelihood to older ages, including the uncorrected age of 7.4 Ma (and older!). The similarity of the posterior distribution to the prior distribution reflects the fact that the measured $^{234}\text{U}/^{238}\text{U}$ activity ratio contains relatively little information. The resulting uncertainties are huge but a correct reflection of our ignorance about the true extent of the disequilibrium in this case.

Finally, changing the $\mu_{4/8}$ ratio to the equilibrium value:

[Figure]

Figure 6: Bayesian alternative to the modified Hoogland example of Figure 3. The posterior distribution is, essentially, identical to the prior distribution.

The similarity of the posterior distribution to the prior means that there is no information in the likelihood function. In other words: the measured $^{234}U/^{238}U$ activity ratio does not tell us anything about the initial disequilibrium. If we truly believe that the sample may have experienced extreme $^{234}U/^{238}U$ disequilibrium, then it is not possible to undo the effects of this disequilibrium on the $^{206}Pb/^{238}U$ clock with the current $^{234}U/^{238}U$ activity ratio.

In conclusion, I think that the authors have two options:

1. If they want to stick with the Monte Carlo approach, then they should modify their code to return an error message when the analytical uncertainty of the $^{234}U/^{238}U$ measurement overlaps with the equilibrium ratio.

2. Alternatively (and preferably), they could replace the ad-hoc Monte Carlo approach with a Bayesian solution along the lines of what I have sketched

in this review. Note, however, that the simple solution presented above is limited in the sense that it only quantifies the uncertainty associated with the disequilibrium correction, and ignores the uncertainty of the actual isochron fit. Doing so would be relatively straightforward using an MCMC algorithm, but I lacked the time to do so for this review.

**References**

Bateman, H. The solution of a system of differential equations occurring in the theory of radio-active transformations. *Proc. Cambridge Phil. Soc., 1908*, 15: 423–427, 1908.

Ludwig, K. R. On the treatment of concordant uranium-lead ages. *Geochimica et Cosmochimica Acta*, 62:665–676, 1998. doi: 10.1016/S0016-7037(98)00059-3.

Pickering, R., Herries, A. I., Woodhead, J. D., Hellstrom, J. C., Green, H. E., Paul, B., Ritzman, T., Strait, D. S., Schoville, B. J., and Hancox, P. J. U–Pb-dated flowstones restrict South African early hominin record to dry climate phases. *Nature*, 565(7738):226, 2019.

Powell, R., Green, E. C., Marillo Sialer, E., and Woodhead, J. Robust isochron calculation. *Geochronology*, 2(2):325–342, 2020.

---

## Author Comment (AC2)

**Response to RC1 on "DQPB: software for calculating disequilibrium U-Pb ages"**

Timothy Pollard et al.

December 8, 2022

We thank Pieter Vermeesch for his thorough review and helpful suggestions regarding an important aspect of the disequilibrium U-Pb age calculations.

The main point raised concerns the accuracy of the error propagation approach employed by DQPB, for samples that have measured $^{234}\text{U}/^{238}\text{U}$ activity ratios (hereafter $[4/8]_m$) which are not clearly resolvable from radioactive equilibrium with respect to measurement uncertainties. We concur that this an important issue, but one that only effects a particular cohort of samples. This issue is not relevant to samples with $[4/8]_m$ values that are clearly resolved from equilibrium, nor ages calculated using an assumed initial $^{234}\text{U}/^{238}\text{U}$ activity ratio (hereafter $[4/8]_i$). For these unaffected samples, we believe that the Monte Carlo approach adopted is appropriate and accurate. The Monte Carlo approach is also preferable to analytical error propagation approaches based on a linear approximation for samples with large regression fitting errors, because such samples can have highly asymmetric age uncertainties.

The review correctly identifies that the Monte Carlo error propagation approach for samples with a $[4/8]_m$ value that is not clearly resolvable from radioactive equilibrium can produce a significant number of failed Monte Carlo iterations. This can occur even if ages or $[4/8]_i$ solutions are not independently constrained to positive values, since some iterations may not have a convergent age solution, and results in unreliable age uncertainty estimates for these samples. Although, DQPB reports the number of failed Monte Carlo iterations, we agree that this provides an insufficient warning to users that age uncertainty results are potentially unreliable.

To address this issue, we have implemented two checks in the software to verify that the data are suitable for the Monte Carlo uncertainty propagation. The first check ensures that the $[4/8]_m$ input value is analytically resolvable from equilibrium with 95% confidence. Where this criterion is not met, a warning is displayed to the user, and the Monte Carlo simulation does not proceed. The age is still reported, but the uncertainties are listed as undefined. The second check, which is performed after a Monte Carlo simulation is completed, verifies

that a minimum number of iterations were successful (the default value is set to 97.5 %). Where this second criterion is not met, the software displays a warning that Monte Carlo simulation results may be unreliable and should not be used.

We have updated the manuscript to include a brief discussion of this issue and outline the limitation of the software in handling such samples.

The Bayesian approach suggested by the reviewer is an interesting idea, but we would argue that it is only really compelling when a well-informed prior is available rather than as a general go-to approach. In the example provided by the reviewer, we believe that the main advantage of a Bayesian approach, namely the ability to include prior information, is greatly underutilised. In this example case, we would argue that the prior should be established based on the distribution of $[4/8]_i$ for other U-Pb determinations from the same cave site which are clearly resolved from equilibrium. Further information may be obtained via the approach of Engel et al. (2019) for samples that are analytically unresolvable from equilibrium.

We address the reviewer's other specific comments below.

Comment: 'DQPB's online documentation is detailed and extensive, but only covers the GUI. It would be useful if it also covered the command-line API. Not only would this allow Linux users to access DQPB, but it would also benefit power users on other operating systems.'

Response: We made a deliberate decision to initially focus on developing documentation for the GUI version of the software as we believed it would be much more widely used. We have now compiled documentation for the pure Python version as well, but apologise for the delay with this.

Comment: 'One slightly annoying issue is that DQPB overprints the data with the results. This problem probably only occurs on Windows (DQPB's lead developer appears to use Mac OS) and should be easy to fix.'

Response: DQPB allows users to choose where the results will be printed within an Excel worksheet, so it will only overprint pre-existing data if an inappropriate output location is selected. However, we agree that this behaviour can sometimes lead to frustration and have updated the software to display a warning if data are going to be overprinted. In this case, we also offer an option to change the print location before proceeding.

Comment: 'I tested DQPB on a number of samples and got similar results to IsoplotR. This is not surprising given that the two programs use, essentially, the same equations, although IsoplotR casts them in a matrix form.'

Response: We greatly appreciate the time taken by the reviewer to test the

software.

We believe that the matrix exponential-based age equations are equivalent to the Bateman form presented in the manuscript, so the two approaches will yield identical results all else being equal. The equivalence between these two formulations can be shown by expanding out the matrix exponential product

$$\boldsymbol{n} = \mathbf{Q}e^{\boldsymbol{\Lambda} t}\mathbf{Q}^{-1}\boldsymbol{n}_i$$

which yields a column vector, $\boldsymbol{n}$, containing nuclide abundances at age $t$. For the purposes of U-Pb age calculation, we are only interested in the last element of $\boldsymbol{n}$, which for the $^{238}$U-$^{206}$Pb decay series can be expressed after some minor algebraic manipulation as

$$\frac{^{206}\text{Pb}^*}{^{238}\text{U}} = e^{\lambda_{238}t}\left(q_{inv}^{11}e^{-\lambda_{238}t} + q_{inv}^{21}e^{-\lambda_{234}t} + q_{inv}^{31}e^{-\lambda_{230}t} + q_{inv}^{41}e^{-\lambda_{226}t} + 1\right)$$

$$+ \left[\frac{^{234}\text{U}}{^{238}\text{U}}\right]_i \frac{\lambda_{238}}{\lambda_{234}}e^{\lambda_{238}t}\left(q_{inv}^{22}e^{-\lambda_{234}t} + q_{inv}^{32}e^{-\lambda_{230}t} + q_{inv}^{42}e^{-\lambda_{226}t} + 1\right)$$

$$+ \left[\frac{^{230}\text{Th}}{^{238}\text{U}}\right]_i \frac{\lambda_{238}}{\lambda_{230}}e^{\lambda_{238}t}\left(q_{inv}^{33}e^{-\lambda_{230}t} + q_{inv}^{43}e^{-\lambda_{226}t} + 1\right)$$

$$+ \left[\frac{^{226}\text{Ra}}{^{238}\text{U}}\right]_i \frac{\lambda_{238}}{\lambda_{226}}e^{\lambda_{238}t}\left(1 - e^{-\lambda_{226}t}\right)$$

whereby $q_{inv}^{ij}$ are the elements that populate the matrix $\mathbf{Q}^{-1}$, and square brackets denote activity ratios. This equation is strictly equivalent to Eq. (1) in the manuscript, since $q_{inv}^{11}$, $q_{inv}^{21}$, $q_{inv}^{31}$, $q_{inv}^{41}$, are equal to Bateman coefficients $c_1$, $c_2$, $c_3$, $c_4$, $q_{inv}^{22}$, $q_{inv}^{32}$, $q_{inv}^{42}$ are equal to $h_1$, $h_2$, $h_3$, and $q_{inv}^{33}$, $q_{inv}^{43}$ are equal to $p_1$, $p_2$. We would argue that calculating Pb*/U ratios by compiling and multiplying out the full matrix product is unnecessary for the purpose of age calculation, and that the more direct form above is preferable in most, if not all, cases.

Comment: 'The manuscript is a bit dismissive of IsoplotR's disequilibrium corrections, even though these are more extensive than DQPB's current capabilities and include 3-dimensional 'Total U-Pb' isochron regression and Ludwig (1998)-style error propagation, neither of which are implemented in DQPB.'

Response: It wasn't our intention to be dismissive of IsoplotR's capabilities in this regard, but rather point out that they are not documented in the peer-reviewed literature, and without digging into the source code, it is not clear exactly what has been implemented. We have modified the manuscript to make this point clearer.

In response to the second point, it is our view that incorporation of the "Total U-Pb" regression approach is of limited use here if it does not propagate disequilibrium correction uncertainties. These can be a substantial source of age uncertainty, especially for samples with small isochron fitting uncertainties

and/or large activity ratio uncertainties. Is it not possible to incorporate these using an equivalent approach to decay constant uncertainties in Ludwig (1998)?

Comment: 'Since DQPB does not work on my computer, I implemented my own version of this algorithm, using R and IsoplotR. The only major difference between my code and DQPB is that it does not sample the $[4/8]_m$-distribution randomly, but uses a targeted approach to sample $[4/8]_m$ as a sequence of regularly spaced normal quantiles. This has two advantages. First, it requires orders of magnitude fewer iterations (50 vs. 30,000). Second, it produces a deterministic result, unlike the Monte Carlo approach, whose results depend on the seed of a random number generator.'

Response: We believe the advantages of this approach are overstated for software that isn't computationally constrained in the way that an online application is. For typical datasets, running 30,000 Monte Carlo iterations on modern computer hardware is typically completed in a few seconds, and the indeterminate nature of Monte Carlo uncertainties is of no practical significance provided a sufficient number of iterations are performed.

**References**

Engel, J., Woodhead, J., Hellstrom, J., Maas, R., Drysdale, R., and Ford, D.: Corrections for Initial Isotopic Disequilibrium in the Speleothem U-Pb Dating Method, Quaternary Geochronology, 54, 101 009, https://doi.org/10.1016/j.quageo.2019.101009, 2019.

Ludwig, K. R.: On the Treatment of Concordant Uranium-Lead Ages, Geochimica et Cosmochimica Acta, 62, 665–676, https://doi.org/10.1016/S0016-7037(98)00059-3, 1998.

---

## Author Response (AR1)

**Reply to the Associate Editor's initial decision on: 'DQPB: software for calculating disequilibrium U-Pb ages'**

**Timothy Pollard et al.**

**February 3, 2023**

We thank Noah McLean for his careful reading of the manuscript and helpful suggestions. We are now pleased to submit a revised version of the manuscript that addresses the Reviewers' comments as discussed in our previous reply documents, together with the Associate Editor's comments as outlined below.

Comment: "An example, perhaps with synthetic data, illustrating where and how DQPB's algorithms break down with user input of this sort would be informative, but it also risks cluttering an already technical paper. I suggest (but don't insist on) adding this to the text if you can find a place for it, to an appendix if you can't, or to the well-developed online documentation if you don't feel it belongs in the manuscript."

Response: In our view, the cases in which the Monte Carlo algorithms fail to deliver reliable results are relatively easy to explain in text. Therefore, we believe that the benefits of including a detailed example with synthetic data in the manuscript (in terms of greater clarity) are outweighed by the disadvantages in terms of increased cluttering. For this reason, we have included a discussion of this issue in the manuscript, but left examples with synthetic data to the documentation.

Comment: "I think the Monte Carlo approach works fine here, and note that DQPB has a setting to adjust the number MC samples generated. There is an error in the mean calculated from Monte Carlo samples that scales with $\sigma/n$. That's 0.0058 of $\sigma$ for 30,000 MC trials, $1/\sqrt{n}$ which would affect the second significant digit of many DQPB-calculated uncertainties and means rounded to the same decimal place. Perhaps it's worth mentioning that $n$ should be increased when calculating dates for publication or when comparing DQPB's output with other calculations."

Response: We have added further discussion of Monte Carlo calculations to the manuscript to ensure that users are aware that the accuracy of Monte Carlo calculated uncertainties scale with the number of trials, $n$. We also suggest that $n$ should be increased when computing final age uncertainties.

Comment: "Line 29 – Igneous minerals are crystallized, not deposited (at least, in this context)."

We have changed the terminology accordingly.

Comment: "Line 31 – Following on the clarifications suggested by Ickert, monazite doesn't incorporate "an initial excess of Th" but instead an initial $^{230}Th/^{238}U$ in excess of the ratio in the melt, which is usually assumed to be at equilibrium with respect to the top of the $^{238}U$ decay chain in U-Pb geochronology."

We have made the language more exact in the revised manuscript.

Comment: "Figure 2 caption – The $^{207}Pb$ age described here and elsewhere is, to me, a model age. If this nomenclature has entered the literature and you're set on using it, I think that's ok. But

calling this a model age might help others make a connection to a relevant, more broadly applied concept. "

The term 'modified $^{207}$Pb age' has entered the literature to some extent, however, we agree that this term is not entirely satisfactory and do not insist on using it. To the best of our knowledge, Sakata (2018) introduced the term 'modified $^{207}$Pb age', because the approach is equivalent to the '$^{207}$Pb-corrected' approach used by SIMS analysts (Williams 1998; Ludwig, 2009), but involves intersection with a 'modified' (i.e., disequilibrium) concordia. Arguably, using the term 'modified' does little to clarify things here. We propose instead using the term '$^{207}$Pb-corrected age' and adding a reference to this SIMS literature, where this kind of approach is more widely used.

Comment: "Line 147 – The Pb isotope ratios in question don't necessarily come from Pb-rich phases, at least I don't think of K-feldspars as Pb-rich. I think you're looking for low $^{238}$U/$^{204}$Pb (aka $\mu$) here."

We have re-phrased this sentence accordingly.

Comment: "Line 154 – The term $D_{Th/U}$ is missing at the end of the line, before 'varies across...' "

This has been corrected.

Comment: "Line 155 – The mineral grains need not be coeval or even cogenetic to do a disequilibrium correction as described here. The Rioux et al. (2012) grains were not assumed coeval, but instead interpreted to have some real spread in age."

It wasn't our intention here to suggest that suites of mineral grains need be coeval or cogenetic to compute disequilibrium corrected ages in this way, but rather state that these assumptions are usually required to compute a meaningful weighted average. However, even if a weighted average is computed, we acknowledge that there are exceptions to the coeval requirement where non-analytical spread in crystallisation ages can confidently be assumed to conform to a particular probability distribution (as in Rioux et al., 2012). We have modified the manuscript to ensure that the text is not interpreted as suggesting that the mineral grains were assumed to be coeval by Rioux et al., (2012).

Comment: "Line 156 – The $^{232}$Th/$^{238}$U is directly measured, in the sense that a $^{232}$Th beam is measured, in many LA-ICPMS U-Pb studies. The $^{232}$Th/$^{238}$U is not directly measured in the vast majority of ID-TIMS studies, and instead the $^{232}$Th is back-calculated from an estimated age and the radiogenic $^{208}$Pb (e.g., Schmitz and Schoene, 2007). I'm not certain why the assumption about radiogenic $^{208}$Pb needs to be made to estimate Th/U$_{min}$ when there is a direct $^{232}$Th/$^{238}$U measurement, unless it's to justify non back-calculating the initial $^{232}$Th/$^{238}$U. "

We accept that this assumption is unsatisfactory for TIMS analyses and have modified the software accordingly. When computing Pb*/U or $^{207}$Pb-corrected ages assuming a constant Th/U$_{melt}$ value, users may now input either a measured $^{232}$Th/$^{238}$U ratio for each aliquot (e.g., in LA-ICPMS and SIMS analyses where a $^{232}$Th beam is measured), or input a measured radiogenic $^{208}$Pb/$^{206}$Pb ratio for each aliquot, from which $^{232}$Th/$^{238}$U may be inferred using an iterative approach (e.g., in TIMS analyses). The algorithm has also been modified so that it no longer makes any assumptions regarding in-growth of radiogenic $^{208}$Pb .

Comment: "Line 168 – Variations in Th and U partitioning behaviour may constitute a systematic component of error, as suggested here, but it's purely systematic only if you assume that there is no mineral-to-mineral variation in this behaviour. Different DTh/U could explain, for instance, the spread in coeval zircon $^{238}$U/$^{232}$Th ratios measured for $^{238}$U- $^{230}$Th dating (e.g., Cooper and Reid, 2008, and references therein). Propagating $D_{Th/U}$ uncertainty as purely systematic does not account for scatter derived from this variation, or alternately/additionally differences in Th/U in a compositionally heterogeneous magmatic system. However, there is often significant systematic uncertainty in the mean of $D_{Th/U}$ or the magma Th/U. Including this source of systematic uncertainty as correlated age uncertainties is commendable. "

The software offers two different approaches to accounting for disequilibrium in Pb*/U and $^{207}$Pb-corrected ages. The first of these assumes that the ratio of mineral–melt partition coefficients (i.e., $D_{Th/U}$) is constant for all mineral grains, and is known a priori within some uncertainty. Applying this approach allows for the possibility that Th/U of the magma is heterogenous. The second approach assumes that Th/U of the melt is homogenous, but mineral-melt partitioning may vary. We acknowledge that these approaches are based on ideal sets of assumptions, and that, in reality, it is possible for there to be variability in the mineral–melt partitioning behaviour and heterogeneity in the melt composition at the same time. However, we believe that the limitations of these two ideal cases are widely understood and discussed within the literature (e.g., Rioux et al. 2012, Guillong et al., 2014.; Kasbolm and Schoene 2018, etc.). Therefore, we believe it is best left up to the user to decide which approach (if either) is most relevant to their particular use case. We have re-written this part of the manuscript in an effort to make the distinctions between these two approaches, and their inherent assumptions, clearer.

Comment: "Line 179 – Reference the definitions of $F$ and $G$ above. "

Done.

Comment: "Figure 3 Plots – Are the uncertainties in the text boxes for the plots $\pm 1\sigma$, $\pm 2\sigma$, or 95% CIs?"

Uncertainties in the text boxes are 95% confidence intervals. We have added some text to the results box to make this clearer.

Comment: "Figure 3 Plots – For the y-intercept, it would be more readable to express the result without scientific notation, as $0.8137 \pm 0.0015\ldots$. In the caption, please mention that s is the spine width and reference the appropriate section of the text."

We have made these changes.

Comment: "I appreciate that (a) does not show the oldest reaches of the concordia curve, which is meaningless in this application but is often plotted anyway."

DQPB outputs two separate graphs for concordia intercept ages because it can be difficult to properly show both the regression fit and the concordia intercept at an appropriate scale on the same plot. The software has options to include the concordia intercept in plot (a), but we believe it is better to show the default outputs of the software in the manuscript examples.

Comment: "Section 4? You might usefully subdivide Section 4 into two sections for 'classical' and robust fitting, then reference 4.2"

We have implemented this suggestion in the revised manuscript.

Comment: "Figure 3 Caption – Change Mid- to Middle. The word ellipses need not be in quotes."

Done.

Comment: "Figure 3 Caption – What is the confidence level of the ellipse representing the MC-ed concordia intercept points?"

The ellipse representing the Monte Carlo concordia intercept points is a 95% confidence ellipse. The confidence level is now stated in the figure caption.

Comment: "Figure 4 – Thanks for emailing the plot for this figure. It shows up when the submitted manuscript is opened in my Chrome browser but not, as reviewer Ickert points out, in Adobe Acrobat Reader. The same comments from Figure 3 apply here — please indicate whether age uncertainties are reported in the plot text boxes as $\pm 1\sigma$, $\pm 2\sigma$ or other. Likewise for the uncertainties in the dates reported in the caption, alongside their MSWD and n, for maximum clarity."

We have made these changes to the figure text box and caption.

Comment: "Section 8.2 – Please indicate that the uncertainties in $D_{Th/U}$ and $D_{Pa/U}$ are propagated as purely systematic uncertainties. You might also explain that this involves the assumption that the $D_{Th/U}$ and $D_{Pa/U}$ are unknown, but are identical for all measured data points, if this is how you are treating the uncertainty propagation (this is my assumption from the dark vs. light blue uncertainty bars in Figure 5b)."

We have added this clarification regarding uncertainty propagation to the figure caption. $D_{Th/U}$ and $D_{Pa/U}$ values were assumed to be $0.2 \pm 0.03$ $(2\sigma)$ and $2.9 \pm 1$ $(2\sigma)$ respectively, and constant for all data points. We have added a further explanation regarding the assigned $D_{Th/U}$ and $D_{Pa/U}$ uncertainties. The dark vs. light uncertainty shading is intended to show the difference between the analytical only uncertainty and the combined random and systemic uncertainty (as in the "Error Bar" text box of Schoene et al., 2013). We have modified the text in the caption to make this clearer.

Comment: "Figure 5 – Please indicate whether age uncertainties are reported in the plot text boxes as $\pm 1\sigma$, $\pm 2\sigma$ or other. The caption states that the dashed blue lines project from the y-axis through the measured data points to the concordia intercept, but it looks like the blue line corresponding to the discordant data point doesn't continue to the concordia curve. In the caption, please indicate in the description of (a) that the dark blue ellipses are the "data ellipses," to distinguish them from the white concordia uncertainty ellipses. Please re-state the uncertainties in $D_{Th/U}$ and $D_{Pa/U}$ from Section 5.2 for maximum clarity, as they are responsible for the yellow band. "

We have specified that the age uncertainties are 95% confidence intervals. We have fixed the figure, so that the lines now project to the concordia line. We have also implemented these other suggested amendments to the figure caption.

Comment: "Line 456 – Change 'which' to 'that' or reword."

Done.

Comment: "Line 463 – Insert the word 'for' after 'framework'. "

Fixed.

Comment: "Line 476 – Change 'spline' to 'spine'."

Fixed.

**References**

Guillong, M., von Quadt, A., Sakata, S., Peytcheva, I., Bachmann, O., 2014. LA-ICP-MS Pb-U dating of young zircons from the Kos–Nisyros volcanic centre, SE aegean arc. Journal of Analytical Atomic Spectrometry 29, 963–970. https://doi.org/10.1039/C4JA00009A

Kasbohm, J., Schoene, B., 2018. Rapid eruption of the Columbia River flood basalt and correlation with the mid-Miocene climate optimum. Science Advances 4. https://doi.org/10.1126/sciadv.aat8223

Ludwig, K.R., 2009. SQUID 2 Rev. 2.50: A user's manual. Berkeley Geochronology Center Special Publication, 5, p.110.

Rioux, M., Johan Lissenberg, C., McLean, N.M., Bowring, S.A., MacLeod, C.J., Hellebrand, E., Shimizu, N., 2012. Protracted timescales of lower crustal growth at the fast-spreading East Pacific Rise. Nature Geosci 5, 275–278. https://doi.org/10.1038/ngeo1378

Sakata, S., 2018. A practical method for calculating the U-Pb age of Quaternary zircon: Correction for common Pb and initial disequilibria. Geochemical Journal 52, 281–286. https://doi.org/10.2343/geochemj.2.0508

Schoene, B., Condon, D.J., Morgan, L., McLean, N., 2013. Precision and accuracy in geochronology. Elements 9, 19–24.

Williams I. S., 1998. U-Th-Pb geochronology by ion microprobe. Applications of Microanalytical Techniques to Understanding Mineralizing Processes (McKibben, M. A., Shanks, W. C., III and Ridley, W. I., eds.), Rev. Econ. Geol. 7, 1–35.

---

## Editor Decision (ED1)

Initial Decision for

`DQPB`: software for calculating disequilibrium U-Pb ages
by Timothy Pollard et al., manuscript *GChron-2022-24*

Noah McLean

December 21, 2022

This is a well-written manuscript describing both U-Pb date calculations with initial isotopic disequilibrium and a useful, open-source software package to quantify and visualize those calculations. The writing is clear and well-organized and the software works as advertised. I agree with the reviewers that this manuscript is well-suited to publication in GChron and suggest minor revisions, as identified by the reviewers and by myself below.

The central concern of the Vermeesch review is handling systems where probability density functions for user-input parameters substantially overlap physically impossible domains. Vermeesch is correct that the results returned by `DQPB` in these scenarios was misleading. The user warning and missing uncertainty described in the authors' reply is a satisfactory remedy, and a description of this behavior in the revised version of the manuscript will benefit readers.

An example, perhaps with synthetic data, illustrating where and how `DQPB`'s algorithms break down with user input of this sort would be informative, but it also risks cluttering an already technical paper. I suggest (but don't insist on) adding this to the text if you can find a place for it, to an appendix if you can't, or to the well-developed online documentation if you don't feel it belongs in the manuscript. I feel strongly that the best geochronology calculations come from a two-way partnership between, on one side, the developers of the software that performs data reduction, error propagation, and visualization, and on the other side, the other geochronologists using that software. On balance, the better the software is documented and its correct use explained, the better the science that comes out the other end. This manuscript and its accompanying software are a nice contribution to the geochronology literature in this respect.

Responses to the rest of Vermeesch's comments, and edits indicated in those responses, all look fine. The final discussion point concerns the overhead of 30,000 Monte Carlo iterations. I think the Monte Carlo approach works fine here, and note that `DQPB` has a setting to adjust the number MC samples generated. There is an error in the mean calculated from Monte Carlo samples that scales with $\sigma/\sqrt{n}$. That's 0.0058 of $\sigma$ for 30,000 MC trials,

which would affect the second significant digit of many `DQPB`-calculated uncertainties and means rounded to the same decimal place. Perhaps it's worth mentioning that $n$ should be increased when calculating dates for publication or when comparing `DQPB`'s output with other calculations.

Responses to the issues raised by reviewer Ickert and revisions proposed by the authors are all appropriate.

**Minor edits and suggestions**

Line 29 – Igneous minerals are crystallized, not deposited (at least, in this context).

Line 31 – Following on the clarifications suggested by Ickert, monazite doesn't incorporate "an initial excess of Th" but instead an initial $^{230}$Th/$^{238}$U in excess of the ratio in the melt, which is usually assumed to be at equilibrium with respect to the top of the $^{238}$U decay chain in U-Pb geochronology.

Figure 2 caption – The $^{207}$Pb age described here and elsewhere is, to me, a model age. If this nomenclature has entered the literature and you're set on using it, I think that's ok. But calling this a model age might help others make a connection to a relevant, more broadly applied concept.

Line 147 – The Pb isotope ratios in question don't necessarily come from Pb-rich phases, at least I don't think of K-feldspars as Pb-rich. I think you're looking for low $^{238}$U/$^{204}$Pb (aka $\mu$) here.

Line 154 – The term $D_{Th/U}$ is missing at the end of the line, before "varies across..."

Line 155 – The mineral grains need not be coeval or even cogenetic to do a disequilibrium correction as described here. The Rioux et al. (2012) grains were not assumed coeval, but instead interpreted to have some real spread in age.

Line 156 – The $^{232}$Th/$^{238}$U is directly measured, in the sense that a $^{232}$Th beam is measured, in many LA-ICPMS U-Pb studies. The $^{232}$Th/$^{238}$U is not directly measured in the vast majority of ID-TIMS studies, and instead the $^{232}$Th is back-calculated from an estimated age and the radiogenic $^{208}$Pb (e.g., Schmitz and Schoene, 2007). I'm not certain why the assumption about radiogenic $^{208}$Pb needs to be made to estimate $Th/U_{min}$ when there is a direct $^{232}$Th/$^{238}$U measurement, unless it's to justify non back-calculating the initial $^{232}$Th/$^{238}$U.

Line 168 – Variations in Th and U partitioning behavior may constitute a systematic component of error, as suggested here, but it's purely systematic only if you assume that there is no mineral-to-mineral variation in this behavior. Different $D_{Th/U}$ could explain, for instance, the spread in coeval zircon $^{238}$U/$^{232}$Th ratios measured for $^{238}$U-$^{230}$Th dating (e.g., Cooper and Reid, 2008, and references therein). Propagating $D_{Th/U}$ uncertainty as purely systematic does not account for scatter derived from this variation, or alternately/additionally differences in Th/U in a compositionally heterogeneous magmatic system. However, there is often significant systematic uncertainty in the mean of $D_{Th/U}$

or the magma Th/U. Including this source of systematic uncertainty as correlated age uncertainties is commendable.

Line 179 – Reference the definitions of $F$ and $G$ above.

Figure 3 Plots – Are the uncertainties in the text boxes for the plots $\pm1\sigma$, $\pm2\sigma$, or 95% CIs? For the y-intercept, it would be more readable to express the result without scientific notation, as $0.8137 \pm 0.0015$. In the caption, please mention that $s$ is the spine width and reference the appropriate section of the text (Section 4? You might usefully subdivide Section 4 into two sections for 'classical' and robust fitting, then reference 4.2). I appreciate that (a) does not show the oldest reaches of the concordia curve, which is meaningless in this application but is often plotted anyway.

Figure 3 Caption – Change Mid- to Middle. The word ellipses need not be in quotes. What is the confidence level of the ellipse representing the MC-ed concordia intercept points?

Figure 4 – Thanks for emailing the plot for this figure. It shows up when the submitted manuscript is opened in my Chrome browser but not, as reviewer Ickert points out, in Adobe Acrobat Reader. The same comments from Figure 3 apply here — please indicate whether age uncertainties are reported in the plot text boxes as $\pm1\sigma$, $\pm2\sigma$ or other. Likewise for the uncertainties in the dates reported in the caption, alongside their MSWD and $n$, for maximum clarity.

Section 8.2 – Please indicate that the uncertainties in $D_{Th/U}$ and $D_{Pa/U}$ are propagated as purely systematic uncertainties. You might also explain that this involves the assumption that the $D_{Th/U}$ and $D_{Pa/U}$ are unknown, but are identical for all measured data points, if this is how you are treating the uncertainty propagation (this is my assumption from the dark vs. light blue uncertainty bars in Figure 5b).

Figure 5 – Please indicate whether age uncertainties are reported in the plot text boxes as $\pm1\sigma$, $\pm2\sigma$ or other. The caption states that the dashed blue lines project from the y-axis through the measured data points to the concordia intercept, but it looks like the blue line corresponding to the discordant data point doesn't continue to the concordia curve. In the caption, please indicate in the description of (a) that the dark blue ellipses are the "data ellipses," to distinguish them from the white concordia uncertainty ellipses. Please re-state the uncertainties in $D_{Th/U}$ and $D_{Pa/U}$ from Section 5.2 for maximum clarity, as they are responsible for the yellow band.

Line 456 – Change 'which' to 'that' or reword.

Line 463 – Insert the word 'for' after 'framework'.

Line 476 – Change 'spline' to 'spine'.